# Aberrant androgen action in prostatic progenitor cells induces oncogenesis and tumor development through IGF1 and Wnt axes

Won Kyung Kim[1,5], Adam W. Olson [1,5], Jiaqi Mi[1], Jinhui Wang[2], Dong-Hoon Lee[1], Vien Le[1], Alex Hiroto[1], Joseph Aldahl[1], Christian H. Nenninger[1], Alyssa J. Buckley[1], Robert Cardiff[3], Sungyong You [4] & Zijie Sun [1]✉

Androgen/androgen receptor (AR) signaling pathways are essential for prostate tumorigenesis. However, the fundamental mechanisms underlying the AR functioning as a tumor promoter in inducing prostatic oncogenesis still remain elusive. Here, we demonstrate that a subpopulation of prostatic Osr1 (odd skipped-related 1)-lineage cells functions as tumor progenitors in prostate tumorigenesis. Single cell transcriptomic analyses reveal that aberrant AR activation in these cells elevates insulin-like growth factor 1 (IGF1) signaling pathways and initiates oncogenic transformation. Elevating IGF1 signaling further cumulates Wnt/β-catenin pathways in transformed cells to promote prostate tumor development. Correlations between altered androgen, IGF1, and Wnt/β-catenin signaling are also identified in human prostate cancer samples, uncovering a dynamic regulatory loop initiated by the AR through prostate cancer development. Co-inhibition of androgen and Wnt-signaling pathways significantly represses the growth of AR-positive tumor cells in both ex-vivo and in-vivo, implicating co-targeting therapeutic strategies for these pathways to treat advanced prostate cancer.

Prostate cancer is one of the most common malignancies in men and claims about 250,000 deaths annually worldwide[1]. The activation of AR through binding of androgens is essential for prostate tumorigenesis[2,3]. Almost all primary prostate cancer cells express the AR and are dependent on androgens for their oncogenic growth and survival[4]. However, the fundamental mechanisms by which the AR, a steroid hormone receptor, functions as a tumor promoter to initiate prostatic oncogenesis and promote tumor progression still remains elusive. Although androgen deprivation therapy (ADT) directly targeting AR-expressing tumor cells is still the first-line treatment for advanced prostate cancer[5], it eventually fails in most patients who consequently develop castration resistant prostate cancer (CRPC), an incurable disease[2,6]. Therefore, oncogenic AR action in prostate cancer development and progression needs to be understood for designing more effective therapies.

Currently, one of the significant challenges in the field of prostate cancer research is still the lack of appropriate animal models that can recapitulate the oncogenic role of AR during prostate cancer initiation, progression, and hormone refractoriness. While a variety of AR transgenic mouse models have been developed for assessing AR action in prostate oncogenesis in the past decades[7], these models all used the mouse probasin (PB) promoter to induce transgenic AR expression and only showed minor pathological changes and failed to develop prostate tumor lesions as observed in human prostate cancer[8,9]. However, activation of the *floxed* human *AR* transgene (*hARtg*) expression in a subpopulation of prostatic progenitors through the

[1]Department of Cancer Biology, Cancer Center and Beckman Research Institute, City of Hope, Duarte, CA, USA. [2]Integrative Genomics Core, Cancer Center and Beckman Research Institute, City of Hope, Duarte, CA, USA. [3]Center for Comparative Medicine, University of California at Davis, Davis, CA, USA. [4]Division of Cancer Biology and Therapeutics, Departments of Surgery, Samuel Oschin Comprehensive Cancer Institute, Cedars-Sinai Medical Center, Los Angeles, CA, USA. [5]These authors contributed equally: Won Kyung Kim, Adam W. Olson. ✉e-mail: zjsun@coh.org

*Osr1* (*odd skipped related1*) promoter-driven *Cre* recombinases developed high-grade prostatic intraepithelial neoplasia (HGPIN) and invasive prostatic adenocarcinoma (PCa) lesions in mice[10]. The expression of transgenic AR was identified specifically in both prostatic atypical and tumor cells of the transgenic mice[10], directly demonstrating a critical role of *hARtg* expression in HGPIN and prostate tumor development. Additionally, development of both HGPIN and prostatic adenocarcinoma lesions in the *AR* transgenic mice regulated through *Osr1-Cre* but not *PB-Cre* implicates the critical cellular properties of prostatic *Osr1*-expressing cells and their derivatives, termed *Osr1*-lineage cells, in initiating oncogenesis and tumor development[10].

In this study, using single-cell RNA sequencing (scRNA-seq), genetic tracing, and other experimental approaches, we uncover an underlying mechanism by which aberrant AR activation elevates IGF1 signaling pathways in prostatic *Osr1*-lineage cells to initiate PIN formation. Subsequently, activated IGF1 signaling further cumulates Wnt/β-catenin activation in atypical PIN cells to promote tumor development. The correlations in co-occurrences of altered androgen, IGF1, and Wnt/β-catenin signaling pathways were further identified in human prostate cancer samples. Co-inhibition of androgen and Wnt signaling pathways showed more potent repressive effects on the growth of *hARtg*+ prostate tumor cells in ex-vivo and xenograft models than either reagent alone, elucidating the promotional role of Wnt activation in *hARtg*-induced prostate tumorigenesis. These findings uncover a dynamic regulatory loop initiated by aberrant AR expression in prostatic progenitors through altering IGF1 and Wnt signaling pathways in inducing prostate oncogenic transformation and tumor development. They also provide scientific evidence leading to the development of future therapeutic strategies by co-targeting these signaling pathways for achieving better clinical outcomes for advanced prostate cancer.

## Results

### Prostatic *Osr1*-lineage cells possess progenitor properties in expansion of prostatic epithelia through embryonic and pubescent development

Development of HGPIN and prostatic adenocarcinoma lesions in *Osr1-Cre*-driven *hARtg* transgenic mice but not in previous AR transgenic models regulated by the PB promoter implicates the critical role of prostatic *Osr1*-lineage cells in prostate tumorigenesis[10]. To assess the cellular properties of prostatic *Osr1*-lineage cells, we examined the molecular characterization of these cells using scRNA-seq analyses with urogenital sinus (UGS), and prepubescent and pubertal prostate tissues (Fig. 1a). Approximately 766, 5676, and 8253 cells from embryonic day E18.5 UGS tissues, and postnatal day P14 and P35 prostate tissues were obtained after filtering, respectively (Supplementary Fig. 1), and then merged and clustered after performing cell cycle regression. Both epithelial and stromal cell subsets were identified in the merged Uniform Manifold Approximation and Projection (UMAP) plots (Fig. 1b), aligning similar cell types based on their transcriptomic profiles using Seurat's integrated method[11,12] (Supplementary Fig. 2a–c). These cell subsets also appeared in individual UMAP plots, suggesting comparable cellular properties from UGS and prostate tissue samples (Fig. 1c, top panel). *Osr1* expression appeared in both mesenchymal and epithelial cells at E18.5 UGS samples but significantly decreased in both total cells and total epithelial cells of P14 ($p = 0.004$ and $p = 0.002$, respectively) and P35 ($p = 0.003$ and $p = 0.002$, respectively) samples (Fig. 1c, bottom panel; Supplementary Fig. 2d, e). A significant positive correlation between expression of *Osr1* with prostatic stem/progenitor cell markers including *Itga6*, *Ly6a* and *Tacstd2*[13], as well as *Psca*[14], a prostate cancer stem cell marker, (Spearman $r = 0.3$, 0.31, 0.4, and 0.41, respectively), was identified in urogenital sinus epithelium (UGE) but not urogenital sinus mesenchyme (UGM) cells (Fig. 1d vs. Supplementary Fig. 2f), suggesting the progenitor properties of *Osr1*-lineage cells in prostatic epithelial

development. Co-expression UMAP plots also showed overlaying *Osr1* with the above cellular markers in epithelial cell clusters of E18.5 samples (arrows, Fig. 1e). To assess the differential fates of embryonic prostatic *Osr1*-expressing cells, we traced them through embryonic and postnatal prostate developmental stages using *Osr1-Cre* activated membrane-bound green fluorescent protein (mGFP) expression in *Rosa26* $^{mTmG-LoxP/+}$:*Osr1*$^{Cre/+}$ (*R26* $^{mTmG/+}$:*Osr1*$^{Cre/+}$) reporter mice (Fig. 1f). The expression of mGFP activated by *Osr1-Cre* mainly appeared in urogenital epithelium and prostatic buds at E15.5 and 18.5, respectively, and mGFP+ cells were expanded robustly in the epithelium of prostatic glands at P14, 35, and 56, through and after puberty (Fig. 1g). Co-immunofluorescence analyses (Co-IF) showed very limited Osr1+mGFP+ cells but many more mGFP+ cells in both E15.5 and E18.5 UGS samples (Fig. 1h, i). Robust mGFP+ cells also appear in postnatal prostate epithelia (Fig. 1g), demonstrating the ability of *Osr1*-expressing cells to expand prostatic epithelia through pubertal prostate development. Co-IF analysis of P56 prostate tissues showed the majority of mGFP+ cells co-stained with endogenous AR or CK8 (Fig. 1j; Supplementary Fig. 2g1–h4). Intriguingly, a portion of mGFP+ cells also showed positive staining for CK5 and p63, basal epithelial cell markers, in the above prostate sections (blue arrows, Fig. 1j; Supplementary Fig. 2i1–j4). No overlay between mGFP with vimentin (Vim) or smooth muscle actin (SMA) staining was observed (Fig. 1j; Supplementary Fig. 2k1–l4). These data consistently demonstrate the regulatory role of *Osr1*-lineage cells in prostatic epithelial development and growth. Accordingly, more mGFP+ epithelial cells were observed in prostate tissues, particularly in the anterior and dorsal lobes, isolated from *R26* $^{mTmG/+}$:*Osr1*$^{Cre/+}$ mice at both P28 and P56 days than those from *R26* $^{mTmG/+}$:*PB*$^{Cre/+}$ counterparts (Supplementary Fig. 3a–f4). Moreover, TP63 + mGFP+ basal epithelial cells were detected in prostate tissues of *R26* $^{mTmG/+}$:*Osr1*$^{Cre/+}$ mice but not in those of *R26* $^{mTmG/+}$:*PB* $^{Cre/+}$ mice (yellow or blue arrows, Supplementary Fig. 3g1-1' vs 3g2-2'). Taken together, these data demonstrate that prostatic *Osr1*-lineage cells possess basal progenitor properties and are able to expand prostatic epithelial cell populations during prostate development.

### Aberrant expression of *hARtg* regulates transcriptome of prostatic *Osr1*-lineage cells and induces PIN and prostatic adenocarcinoma development

Development of HGPIN and PCa lesions in *Osr1-Cre*-driven *hARtg* mice suggests the promotional role of *hARtg* in *Osr1*-lineage cells in prostate tumorigenesis[10]. To better understand the oncogenic role of *hARtg* in *Osr1*-lineage cells, we generated *R26* $^{mTmG-LoxP/hAR-LoxP}$:*Osr1*$^{Cre/+}$ (*R26*$^{mTmG/hAR}$:*Osr1*$^{Cre/+}$) mice, in which the expression of *mGFP* and transgenic AR co-occurs through *Osr1-Cre* mediated activation (Fig. 2a). Both HGPIN and prostatic PCa lesions developed in *R26*$^{mTmG/hAR}$:*Osr1*$^{Cre/+}$ mice starting at 4 and 10 months, respectively (Supplementary Table 1) but not in control littermates, which is similar as observed in *R26*$^{hAR/+}$:*Osr1*$^{Cre/+}$ mice[10]. Prostate tissues isolated from 6- and 12-month-old *R26*$^{mTmG/hAR}$:*Osr1*$^{Cre/+}$ mice showed typical HGPIN and PCa lesions (Fig. 2b, c), respectively, which were used for preparing scRNA-seq analyses (see below). A uniform nuclear staining for *hARtg* appeared in both atypical and tumor cells within PIN and PCa lesions (Fig. 2d, e). Co-staining of *hARtg* and mGFP were also revealed in both atypical and tumor cells (Fig. 2f), demonstrating their origin deriving from *Osr1*-expressing cells. Interestingly, while the majority of atypical and tumor cells showed positive staining for both *hARtg* and CK8, a portion of atypical cells showed a clear overlay of *hARtg* and CK5 expression within PIN lesions (Fig. 2f). The expression of both *hARtg* and mGFP in atypical and tumor cells within HGPIN and PCa lesions demonstrate the critical role of transgenic AR in *Osr1*-lineage cells in inducing prostatic oncogenesis and promoting PIN and PCa development.

To gain in-depth mechanistic insight into the oncogenic role of *hARtg*, we performed scRNA-seq analyses using the above pathologically confirmed HGPIN and PCa tissues (see Methods). Both scRNA-seq

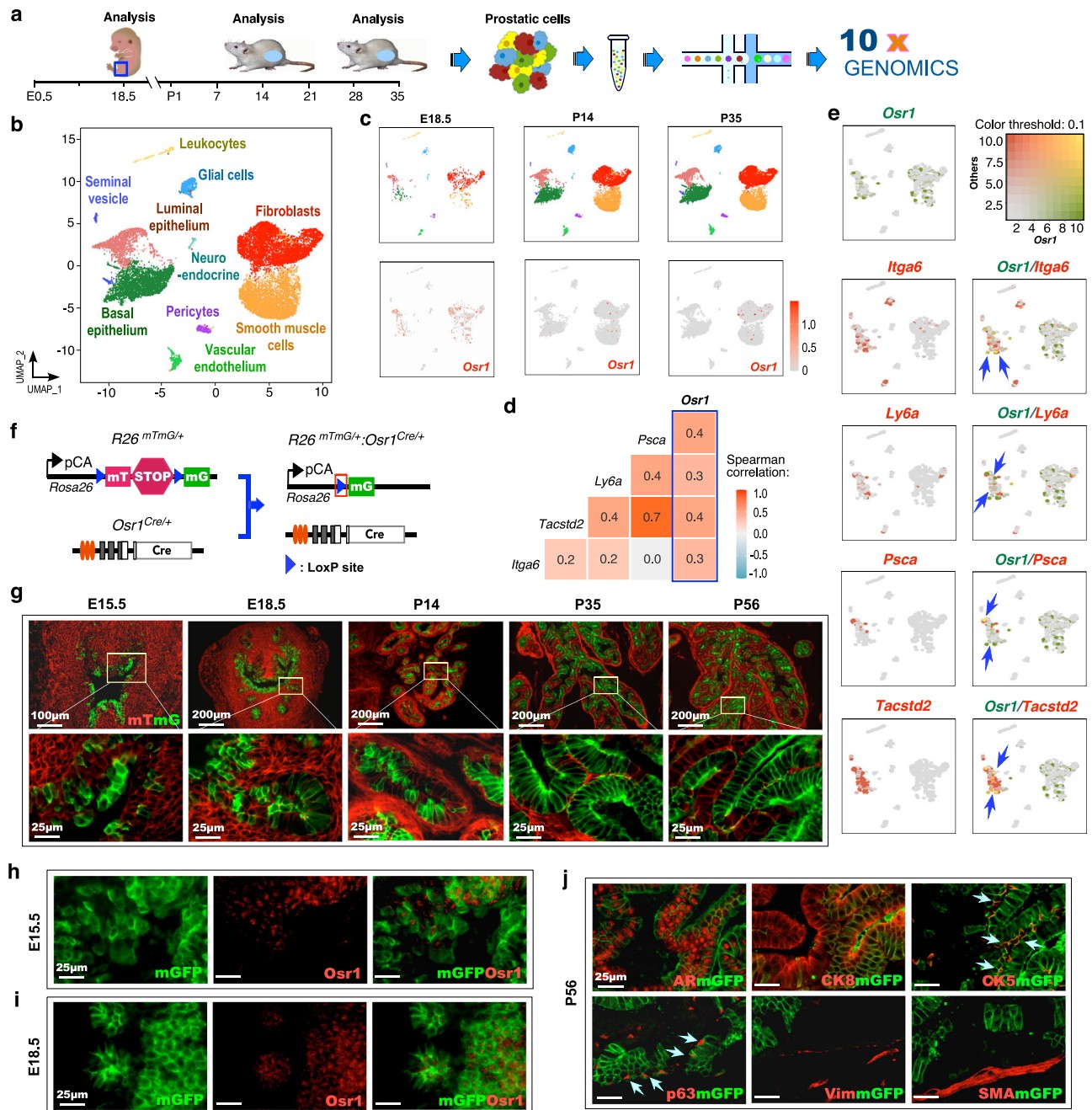

**Fig. 1 | Prostatic *Osr1*-expressing cells possess progenitor properties in expansion of prostatic epithelia through embryonic and pubescent development. a** Schematic of the single cell RNA-sequencing experiment performed. **b** Uniform Manifold Approximation and Projection (UMAP) plot showing the cell type cluster identities based on gene expression patterns. **c** Three individual UMAP plots of single cells at the indicated time points after separation from the original clustering. Gene expression UMAP plots displaying the expression pattern of the *Osr1* gene at the indicated time points. Color intensity indicates the scaled expression level in each cell. **d** Heatmap of pairwise Spearman correlation between the indicated gene expression in epithelial cells from E18.5 male urogenital sinus (UGS). Colors reflect the level of correlation (positive correlation in red and negative correlation in blue); numbers show the correlation coefficient. The blue box highlights the correlation between *Osr1* and other progenitor cell markers. **e** Blended expression UMAP plots displaying the expression of *Osr1* and the indicated genes in the E18.5 male UGS. Blue arrows show the overlay (yellow color) of *Osr1* and the indicated genes in the E18.5 male UGS. **f** Schematics of the *R26^{mTmG/+}* and *Osr1^{Cre/+}* alleles, shown in relation to the mating strategy for this experiment. **g** Representative fluorescence images of mouse UGS or prostate tissues at the indicated timepoints expressing mGFP reporter fluorescence (green) controlled by *Cre* recombination. Scale bars, 100 μm; 200 μm; 25 μm. **h–j** Representative fluorescence images of co-immunofluoresence (IF) staining for the indicated proteins using mouse UGS or prostate tissues at the indicated timepoints. Blue arrows indicate the overlay between mGFP and cytokeratin 5 (CK5) or p63, basal epithelial cell markers **j**. AR, androgen receptor; CK8, cytokeratin 8; Vim, vimentin; SMA, smooth muscle actin. Scale bars, 25 μm. Representative images from three independent experiments with similar results are displayed for each micrograph. See also Supplementary Figs. 1–3.

samples prepared from PIN and PCa tissues underwent multiple steps of filtering following sequencing and alignment to the mm10 reference genome with addition of *mGFP* and *hARtg* sequences[15]. Post-filtering, 6332 or 9412 cells from PIN or PCa samples, with an average of 3751 or 3532 genes and 21,297 or 19,899 UMI counts per cell, respectively, were used in the study (Supplementary Fig. 4a, b). Both PIN and PCa samples were initially visualized individually using UMAP (Fig. 2g), and then merged and clustered, aligning similar cell subsets based on their

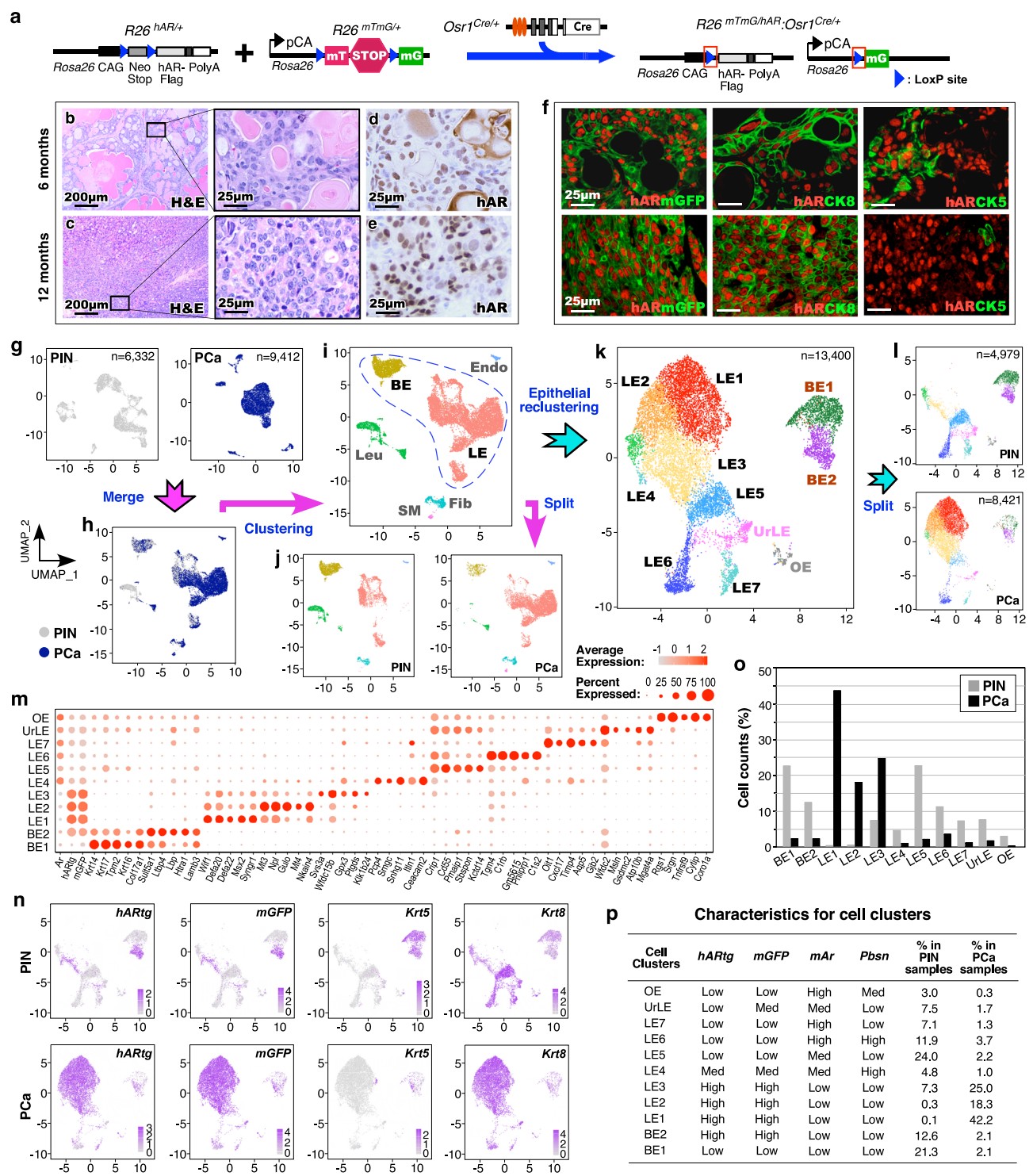

transcriptomic profiles using Seurat's integrated method[11] (Fig. 2h). A total of 6 cell subsets were identified in both merged and individual PIN and PCa plots (Fig. 2i, j), demonstrating comparable cellular properties between these two samples. Specifically, the expression of *hARtg* and *mGFP* are comparable and restricted within the luminal and basal epithelial cell subsets in combined UMAP plots (Supplementary Fig. 4c). Those epithelial cell subsets were further validated with representative epithelial markers (Supplementary Fig. 4d). To gain higher resolution of *hARtg*+ cells in PIN and PCa samples, epithelial cells were separated from other non-epithelial cells, and re-clustered following cell cycle regression (Supplementary Fig. 4e)[16]. Eleven cell clusters were yielded, including two basal, seven luminal, a urethral

epithelial (UrLE)[17], and another epithelial cluster, deemed other epithelia (OE) (Fig. 2k). Five highly expressed genes in each cell cluster were identified to represent their cellular properties (Fig. 2m). Prostatic basal epithelial cells showed high expression of *Krt5*, *Krt14*, and *Trp63*, while luminal cell clusters displayed elevated expression of *Krt18*, *Krt19*, and *Pbsn*, respectively[18]. The expression of *hARtg* and *mGFP* appears mainly in BE2 and LE1-3 clusters, and the expression of endogenous *Ar* is restricted within LE4-7 clusters. All cell clusters appeared in both individual PIN and PCa UMAP plots. However, luminal epithelial cell clusters, LE1 to 3, appeared predominantly in PCa samples, and the rest of the cell clusters showed abundantly in PIN samples (Fig. 2l). UMAP expression plots showed *Krt5* and *Krt8*

**Fig. 2 | Single cell transcriptomic analyses of prostatic intraepithelial neoplasia and adenocarcinoma tissues originating from *Osr1*-expressing cells with *hARtg* expression. a** Schematic of the floxed human androgen receptor (hAR) transgene ($R26^{hAR/+}$) and $R26^{mTmG/+}$ alleles, as well as the corresponding recombined $Osr1^{Cre/+}$ alleles, shown in relation to the mating strategy for this experiment. **b–f** Representative images of hematoxylin-eosin (H&E), immunohistochemistry (IHC), and co-IF staining using the indicated antibodies on adjacent prostate tissue sections from $R26^{mTmG/hAR}:Osr1^{Cre/+}$ mice. Scale bars, 100 μm; 25 μm. **g** Two individual UMAP plots of single cells isolated from prostatic intraepithelial neoplasia (PIN; gray) and prostate adenocarcinoma (PCa; dark blue) tissues, respectively. **h** UMAP visualization of the single cells following integration and clustering of the PIN (gray) and PCa (dark blue) cells. **i** Cell type cluster identities shown on the UMAP of the single cells based on their gene expression profiles. BE, basal epithelium; LE, luminal epithelium; Fib, fibroblasts; SM, smooth muscle cells; Endo, vascular endothelium; Leu, leukocytes. **j** UMAP plot from **i** is split by PIN or PCa cells colored by cell type. **k** UMAP plot of epithelial cells from PIN and PCa tissues. Epithelial cells were sub-clustered, re-clustered and colored by cell cluster. UrLE, urethral epithelium; OE, other epithelium. **l** UMAP plot from **k** is separated by PIN or PCa samples colored by cell type. **m** Dot plot of *Ar*, human AR transgene (*hARtg*), *mGFP*, as well as five cluster-specific genes for each epithelial cell cluster. Dot size indicates the percentage of cells in a cluster expressing each gene; color shows expression level. **n** Gene expression UMAP plots for the indicated genes after separating PIN (top) and PCa (bottom) samples. Color intensity indicates the scaled expression level. **o** Bar chart showing the cell counts within individual clusters from epithelial cells of PIN (gray) or PCa (black) tissues. Source data are provided as a Source Data file. **p** Table summarizing the characteristics for each cell cluster from PIN or PCa samples. Representative images with consistent results from three replicates are shown. See also Supplementary Fig. 4.

expression selectively in basal and luminal epithelial cell clusters of both PIN and PCa samples while *hARtg* and *mGFP* expression was mainly revealed within the BE2 cluster in PIN, and LE1-3 clusters in PCa samples (Fig. 2n). Cell distribution analyses further confirmed luminal clusters (LE1-3) being mostly in PCa samples and other cell clusters being abundant in PIN samples (Fig. 2o). LE1-2 clusters showed relatively high expression of *hARtg* and mGFP but little or no expression of mouse *Ar* and *Pbsn* (Fig. 2p), suggesting their transformed and un-differentiated cellular properties. These data provide high-resolution insight into the cellular properties of prostatic *hARtg*+ atypical and tumor cells derived from *Osr1*-lineage, implicating the direct role of transgenic AR activation in PIN and prostate tumor development in $R26^{mTmG/hAR}:Osr1^{Cre/+}$ mice.

## Transgenic AR expression elevates IGF1 signaling in atypical *Osr1*-lineage basal epithelial cells within PIN lesions

Identifying specific expression of *hARtg* in a subpopulation of atypical basal epithelial cells within BE2 cluster in $R26^{mTmG/hAR}:Osr1^{Cre/+}$ samples (Fig. 2m, n, p) suggests the important role of these *hARtg*+ basal epithelial cells in PIN initiation (Fig. 3a). Additionally, robust mGFP+TP63+ prostatic basal epithelial cells revealed in prostatic tissues of $R26^{mTmG/+}$:*Osr1*$^{Cre/+}$ mice but not in those of $R26^{mTmG/+}:PB^{Cre/+}$ counterparts (Supplementary Fig. 3g1, 2'). Moreover, only $R26^{mTmG/hAR}:Osr1^{Cre/+}$ mice developed HGPIN and PCa lesions but not in AR transgenic mice driven by PB promoters[8–10,19]. These data consistently demonstrate the significance of the *hARtg*+ atypical basal epithelial cells in initiating prostate oncogenesis. To gain direct insight into the regulatory role of *hARtg* in basal epithelial cells, we identified the differentially expressed genes (DEGs) between *hARtg*+ and *hARtg*−basal epithelial cells of PIN and PCa samples using a Wilcoxon Rank Sum test, which showed more than 5% of cells with adjusted *p* values < 0.05 by the Benjamini-Hochberg procedure and average log fold change >0.1 (Fig. 3a, b; Supplementary Data 1). Ingenuity pathway analysis (IPA) of the above DEGs identified significant enrichment in IGF1, Insulin secretion, Wnt/β-catenin, and JAK/STAT signaling pathways (Fig. 3c), which directly regulate prostate tumorigenesis[20]. Specifically, identifying enriched IGF1 and Insulin related signaling pathways in *hARtg*+ basal cells is intriguing. Multiple lines of evidence have shown that IGF1 signaling directly induces and promotes PIN development in both mouse prostate cancer models and human prostate cancers[21–23]. A significant increase in *hARtg* and *Igf1r*, as well as IGF1 signaling downstream genes, *Jak2* and *Mapk13* expression was further identified in *hARtg*+ compared to *hARtg*- basal epithelial cells using box plots (Fig. 3d). A correlation between *hARtg* with *Igf1r* or its downstream targets, *Jak2* and *Mapk13*, was further identified using Spearman gene-gene correlation analysis (Fig. 3e). Quantitative reverse transcription-PCR (qRT-PCR) analyses showed the higher expression of IGF1 signaling effectors, including *Igf1r*, *Jak2*, and *Mapk13*, in PIN tissue samples of $R26^{mTmG/hAR}:Osr1^{Cre/+}$ mice than prostate tissues of $R26^{mTmG/+}:Osr1^{Cre/+}$ controls (Fig. 3f). Using triple-IF analyses, the regulatory role of transgenic AR in activating IGF1R axis was further assessed in PIN tissues of $R26^{mTmG/hAR}:Osr1^{Cre/+}$ mice. Whereas CK14 expression revealed in basal epithelial cells in both abnormal PIN prostatic glands and adjacent normal glands (Fig. 3g), co-expression of mGFP, representing *hARtg*-expressing cells, and IGF1R appeared mainly in atypical cells within PIN areas (Fig. 3g). Overlay of CK14 and mGFP, mGFP and IGF1R, or CK14 and IGF1R appeared selectively in atypical cells within the above PIN lesions (pink boxes and arrows, Fig. 3g) but no or very few cells with the above overlays were observed within normal glandular areas (blue boxes and arrows, Fig. 3g). Triple positive cells of CK14, mGFP, and IGF1R revealed only in atypical cells within abnormal prostatic glands (pink boxes pink arrows, Fig. 3g, right panels), but not in normal glandular cells (blue boxes and arrows, Fig. 3g, right panels). Quantified IGF1R expression in approximately a total of 500 to 800 CK14 + mGFP+ or CK14 + mGFP− cells from five different areas in each sample showed significantly more IGF1R expression in CK14 + mGFP+ cells than CK14 + mGFP− cells in three different experiments with three tissue samples prepared from three different $R26^{mTmG/hAR}:Osr1^{Cre/+}$ mice ($p = 0.00001$) (Fig. 3h).

Identifying aberrant activation of IGF1 signaling pathways specifically in *hARtg*+ atypical basal cells within PIN lesions implicates a regulatory mechanism for *hARtg* in initiating prostatic oncogenesis through the activation of IGF1R axes. Given the AR functioning as a ligand-dependent transcriptional factor, we then performed chromatin immunoprecipitation sequencing (ChIPseq) analysis to examine if *hARtg* regulates IGF1R transcription. A specific enrichment of transgenic AR was identified within the *Igf1r* promoter region on mouse chromosome 7 in the human AR antibody immunoprecipitated samples prepared from PIN tissues of $R26^{mTmG/hAR}:Osr1^{Cre/+}$ mice (Supplementary Fig. 5; Supplementary Data 2). ChIP-quantitative polymerase chain reaction (ChIP-qPCR) analyses further showed the specific occupancy of the transgenic AR in both the promoter and enhancer regions of the *Igf1r* gene with the androgen response element sequences[24], but not in the *Untr4* locus, used as a negative control, in the same immunoprecipitated PIN samples (Fig. 3i, j). Using prostatic organoids derived from $R26^{mTmG/hAR}:Osr1^{Cre/+}$ mice, we also identified the higher expression of *Igf1r* transcripts in samples cultured with dihydrotestosterone (DHT) than those without DHT or treated with a combination of DHT and Enzalutamide (Enz), an antiandrogen (Supplementary Fig. 5c). Taken together, these lines of experimental evidence demonstrate a direct role of *hARtg* in regulating IGF1R expression, implicating an underlying mechanism by which aberrant AR induces PIN development via activating IGF1R signaling.

## Activating Wnt signaling in *hARtg* positive prostatic tumor cells

Our finding of *hARtg* to activate IGF1R axes in atypical basal cells explores a mechanism for AR's oncogenic role in prostate oncogenesis. Prostatic basal epithelial cells have been shown to possess the ability to initiate oncogenic transformation and to further transdifferentiate to luminal tumor cells in the presence of androgens[25–27]. To assess the underlying mechanisms for *hARtg*+ atypical cells to progress to

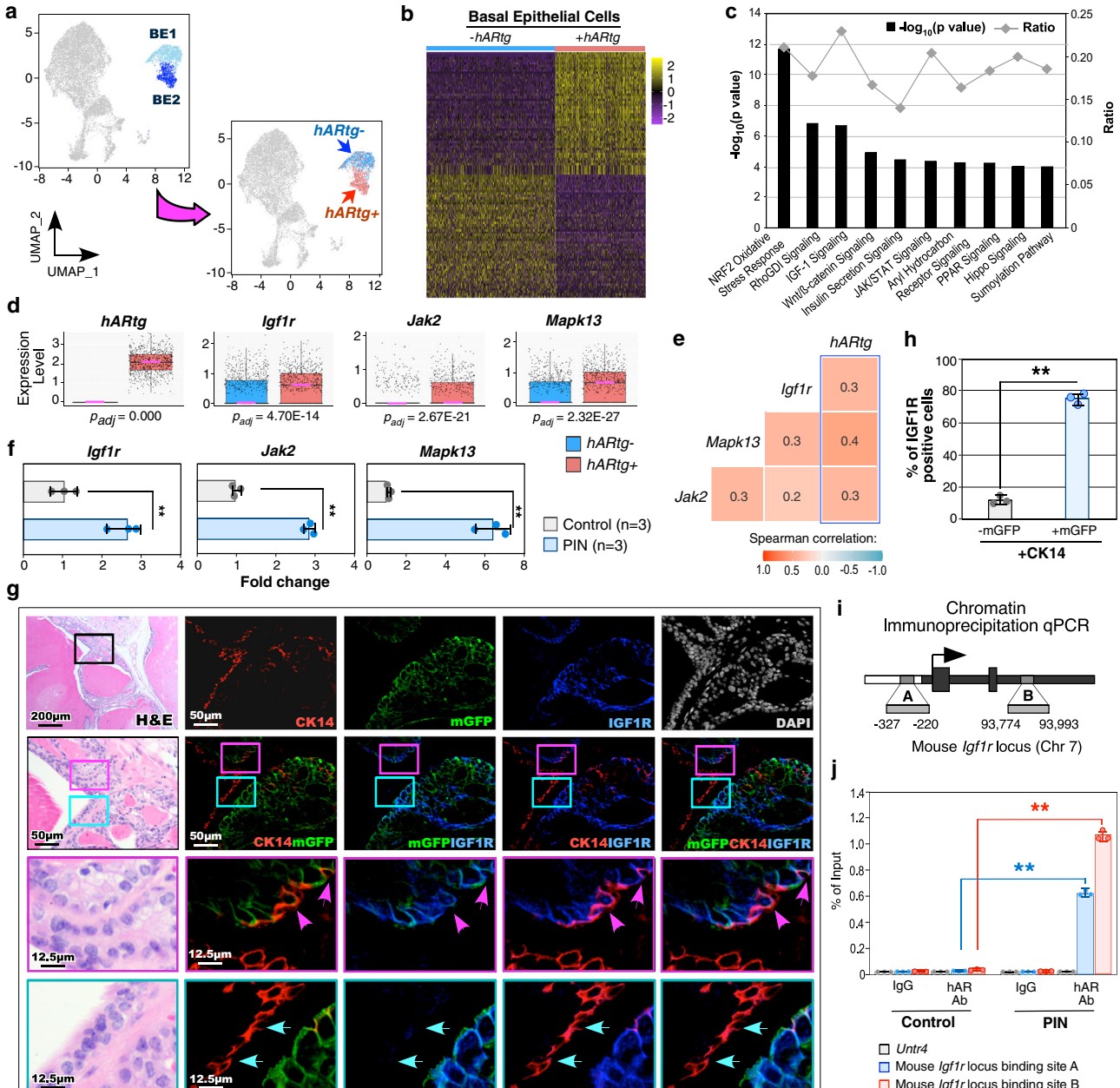

**Fig. 3 | Transgenic AR expression elevates IGF1 signaling in atypical *Osr1*-driven prostatic basal epithelial cells within PIN lesions. a** UMAP plot indicating basal epithelial (BE) cells from PIN and PCa tissues of *R26*^*mTmG/hAR*^:*Osr1*^*Cre/+*^ mice (left), colored by *hARtg* expression (right). **b** Heatmap showing top 50 differentially expressed genes (DEGs) between *hARtg* + and *hARtg*- BE cells. **c** Ingenuity pathway analysis (IPA) pathway analysis of DEGs comparing *hARtg* + and *hARtg*-BE cells. Fisher's exact test (two-sided). Ratio denotes the number of DEGs compared with the total number of genes associated with the canonical pathway. **d** Box-plots representing scaled expression for the indicated genes between *hARtg* + (*n* = 749) and *hARtg*- (*n* = 933) BE cells. Pink lines mark the median; top and bottom lines of boxes indicate the boundaries of the first and third quartiles, respectively; the top and bottom whiskers show the maximum and minimum values, respectively, excluding outliers. Wilcoxon rank-sum test (two-sided) followed by Benjamini-Hochberg correction. **e** Heatmap of pairwise Spearman correlation between indicated gene expression in BE cells. Numbers indicate

correlation coefficient. **f** RT-qPCR analysis of the indicated genes shown as fold change in the indicated tissues. **g** Representative images of H&E and triple-IF staining for the indicated antibodies on adjacent prostate tissues from *R26*^*mTmG/hAR*^:*Osr1*^*Cre/+*^ mice. Pink and blue boxes indicate abnormal and normal glandular areas, respectively, and pink and blue arrows indicate atypical and normal cells, respectively. Nuclei were stained with DAPI. Scale bars, 200 μm; 50 μm; 12.5 μm. **h** Bar chart for the percentage of IGF1R + cells in mGFP− (black) or mGFP + (gray) CK14-expressing BE cells. **i** Diagram showing the mouse *Igf1r* locus containing two AR-binding sites. **j** hAR ChIP-qPCR analysis shown as percent input of *Igf1r* gene on its binding sites A and B in the indicated tissues. In **f**, **h**, and **j**, data are represented as mean ± SD of three biological replicates. Two-sided *t*-test, **p < 0.01. Representative images with consistent results from three replicates are shown. See also Supplementary Fig. 5 and Supplementary Data 1, 2. Source data and the exact *p*-values are provided in the Source Data file.

prostate tumors, we analyzed the transcriptomic changes between LE1-2 tumor cell clusters possessing enriched *hARtg*+ tumor cells and normal LE5-7 clusters containing *hARtg*- cells (Fig. 4a, b). 1834 up-regulated and 3156 down-regulated DEGs were identified (Fig. 4b; Supplementary Data 3). A significant increase in the expression of Wnt/β-catenin target genes, such as *Tcf4*, *Ccnd1*, *Axin2* and *Lgr5*, revealed in LE1-2 in comparison to LE5-7 clusters using either box plots (Fig. 4c) or UMAP expression plots (Fig. 4d). Strong correlations between *hARtg*

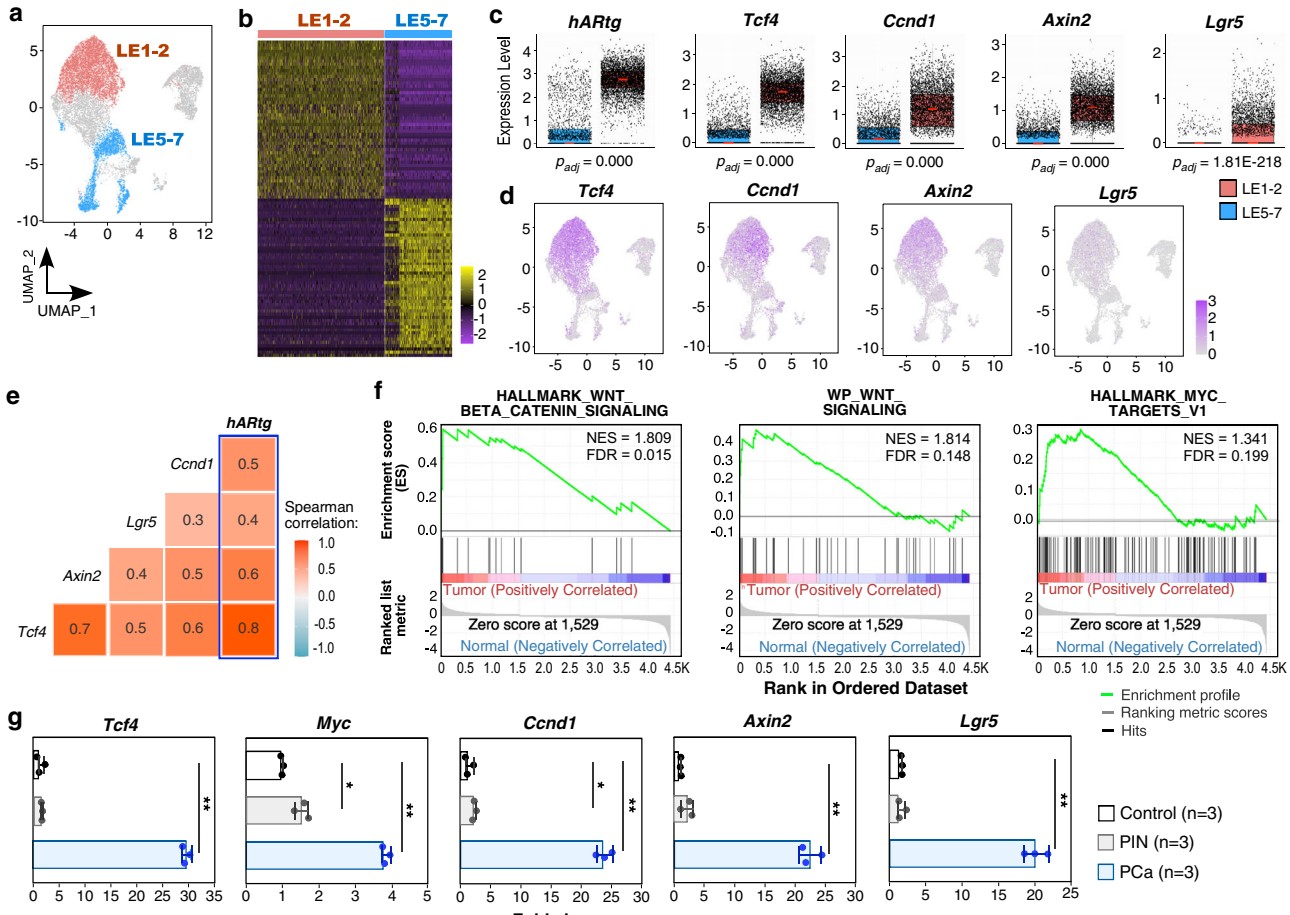

**Fig. 4 | Activating Wnt signaling pathways in *hARtg* positive prostatic tumor cells. a** UMAP plot highlighting LE1-2 and LE5-7 clusters identified as *hARtg*+ tumor cell clusters and *hARtg*-normal cell clusters, respectively. **b** Heatmap showing top 50 differentially expressed genes between LE1-2 (*n* = 5122) and LE5-7 (*n* = 2743) cell clusters. Yellow and purple indicate high and low expression, respectively. **c** Box plots representing scaled expression data for *hARtg* and Wnt downstream target genes applied to LE1-2 and LE5-7 cells. Pink lines mark the median; top and bottom lines of boxes indicate the boundaries of the first and third quartiles, respectively; the top and bottom whiskers show the maximum and minimum values, respectively, excluding outliers. *P*-values were computed using Wilcoxon rank-sum test (two-sided) and adjusted using Benjamini-Hochberg correction. **d** Gene expression UMAP plots displaying the expression patterns for the indicated Wnt downstream genes. Color intensity indicates the scaled expression level in each cell. **e** Heatmap of pairwise Spearman correlation between the indicated gene expression in LE1-2 and LE5-7 cells. Colors reflect the level of correlation (positive correlation in red and negative correlation in blue); numbers show the correlation coefficient. The blue box highlights the correlation between *hARtg* and other Wnt downstream genes. **f** GSEA enrichment plots highlighting the positive enrichment of Wnt signaling pathway with multiple gene sets comparing LE1-2 to LE5-7 cells. **g** RT-qPCR analysis of the indicated genes shown as fold change in normal prostate tissues from *R26^mTmG/+^:Osr1^Cre/+^* control mice and PIN or PCa tissues from *R26^mTmG/hAR^:Osr1^Cre/+^* mice. Data are represented as mean ± SD of three biological replicates. Two-sided *t*-test for PIN or PCa versus control, *$p < 0.05$, **$p < 0.01$. See also Supplementary Data 3. Source data and the exact *p*-values are provided in the Source Data file.

and these Wnt downstream target genes were further confirmed in LE1-2 and LE5-7 cell clusters (Fig. 4e). GSEA using pre-ranked gene lists of LE1-2 versus LE5-7 further showed a significant enrichment in both Wnt and Myc target signaling pathways (Fig. 4f). Increased expression of Wnt/β-catenin downstream target genes, including *Tcf4*, *c-Myc*, *Ccnd1*, *Axin2*, and *Lgr5*, was also identified in RNA samples isolated from pathologically confirmed PCa samples in comparison to PIN and prostate tissues from age and sex-matched wild type (WT) mice using qRT-PCR (Fig. 4g). These data consistently demonstrate the activation of Wnt/β-catenin signaling pathways in *hARtg*+ prostate tumor cells of *R26^mTmG/hAR^:Osr1^Cre/+^* mice.

### Activation of transgenic AR expression in *Osr1*-lineage cells upregulates Wnt signaling pathways in prostate tumor tissues
To gain more in-depth insight into transcriptomic changes in *hARtg* regulated prostatic oncogenesis, we performed bulk RNA-sequencing (RNA-seq) analyses using RNA samples isolated from microscopically confirmed PIN or PCa tissues with more than 80% atypical or tumor cells, respectively, from *R26^mTmG/hAR^:Osr1^Cre/+^* mice,

as well as prostate tissues from sex and age-matched WT mice. Using a median fold difference test (see the Methods), we identified 359, 2970 and 1887 DEGs with adjusted *p* value < 0.05 and fold change ≥ 2 by comparing RNA-seq samples between PIN or PCa versus WT controls, and PCa versus PIN samples, respectively (Fig. 5a; Supplementary Data 4–6). Crossing analyses from the above datasets identified common DEGs between each of the two or three groups (Fig. 5b). A group of common DEGs (*n* = 198) overlapping between PIN or PCa versus WT samples was identified (Fig. 5b; Supplementary Data 4–6). GSEA using pre-ranked DEG lists from PIN or PCa versus WT samples revealed four common significantly enriched signaling pathways, including the mitotic spindle, G₂/M checkpoint, IL6-JAK-STAT3 signaling, and estrogen response pathways (red line, Fig. 5c). Much higher enrichment scores (ES) appeared in the samples of PCa versus WT than those of PIN versus WT, suggesting the promotional roles of these signaling pathways during disease progression. Interestingly, the significant enrichment of Wnt/β-catenin and Myc targets V1 signaling pathways only appeared in the gene list of PCa versus WT samples (Fig. 5c).

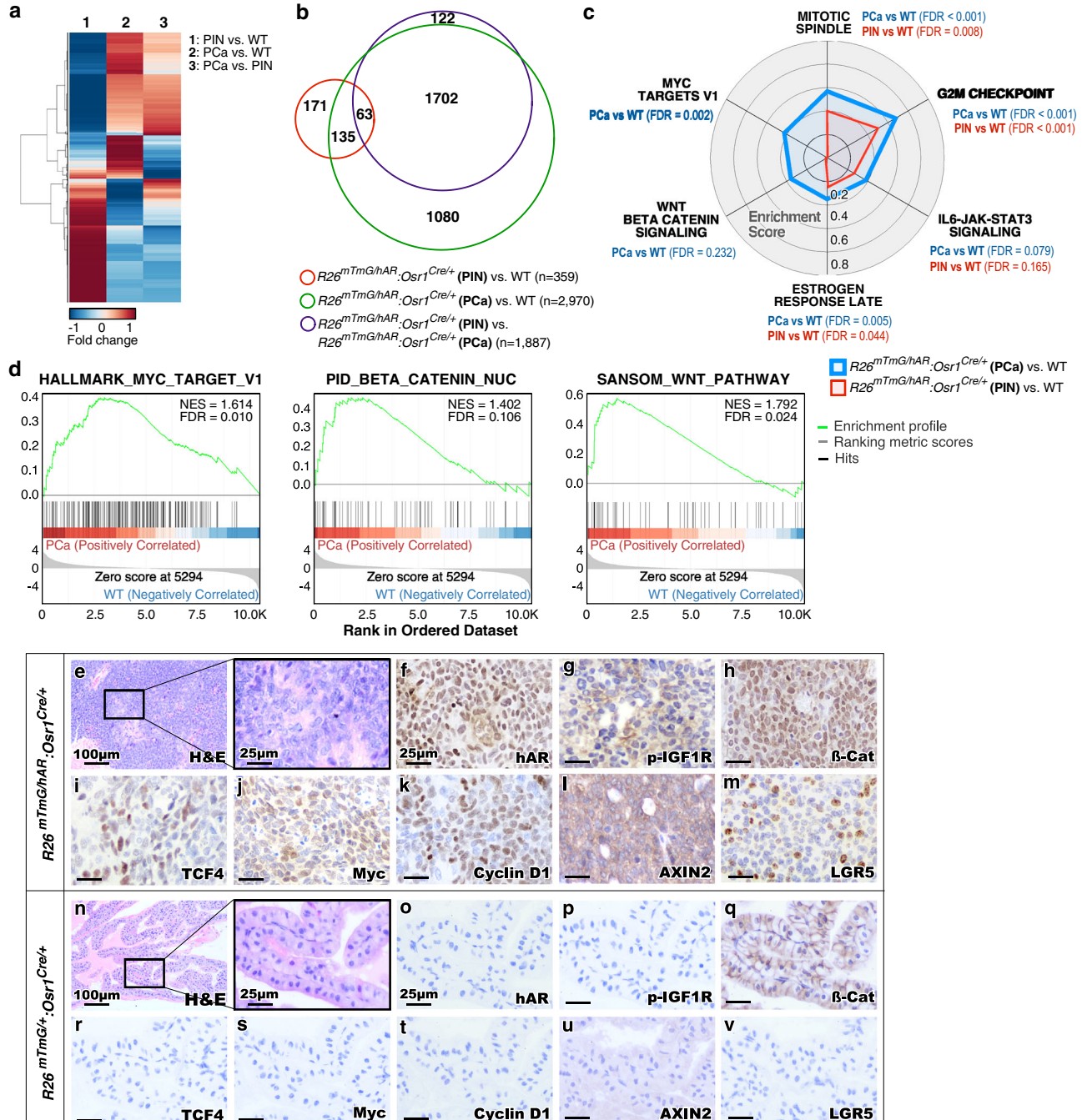

**Fig. 5 | Activation of transgenic AR expression in Osr1-expressing cells upregulates Wnt signaling pathways in prostate tumor tissues. a** Heatmap showing the expression patterns of differentially expressed genes from the three comparisons of bulk RNAseq data from normal prostate tissues from $R26^{mTmG/+}:Osr1^{Cre/+}$ wild type (WT) mice and PIN or PCa tissues from $R26^{mTmG/hAR}:Osr1^{Cre/+}$ mice as indicated in the figure. Red and blue colors indicate up- and down-regulation, respectively. **b** Venn diagram depicting the number of differentially expressed genes from the different comparisons as labeled. **c** Radar chart of the enrichment score displaying differential enrichment of $R26^{mTmG/hAR}:Osr1^{Cre/+}$ (PIN or PCa) in comparison with wild type (WT) control samples. Enrichment score and false discovery rate (FDR) were calculated using a hypergeometric test from GSEA with DEGs (|Log2 fold-change| >1 and adjusted *p* value < 0.05). *P* values were adjusted for multiple testing with Benjamini-Hochberg correction. Red and blue lines indicate enriched hallmark gene sets in PIN and PCa tissues, respectively. d GSEA enrichment plots of pre-ranked gene list from differentially expressed genes comparing PCa to wild type samples, highlighting positive enrichment of Wnt signaling pathway with multiple gene sets. e-v Representative images of H&E and IHC staining using the indicated antibodies on adjacent PCa tissue sections from $R26^{mTmG/hAR}:Osr1^{Cre/+}$ mice and wild type mice. Scale bars, 100 μm; 25 μm. Images representative of consistent results from three independent experiments are shown. See also Supplementary Data 4–6.

GSEA with a pre-ranked DEG list by comparing PCa to WT samples further showed the significant enrichment (FDR < 0.25) in both Wnt/β-catenin signaling and Myc targets (Fig. 5d). IHC analyses using adjacent PCa tissue sections revealed positive nuclear staining for transgenic AR and β-catenin, as well as positive staining for phosphorylated IGF1R and for β-catenin downstream targets, including *TCF4*, *c-Myc*, *Cyclin D1*, *AXIN2*, and *Lgr5* (Fig. 5f–m). In contrast, no specific staining was detected in normal prostate tissues isolated from age- and sex-matched $R26^{mTmG/+}:Osr1^{Cre/+}$ controls (Fig. 5o–v). These data consistently demonstrate aberrant

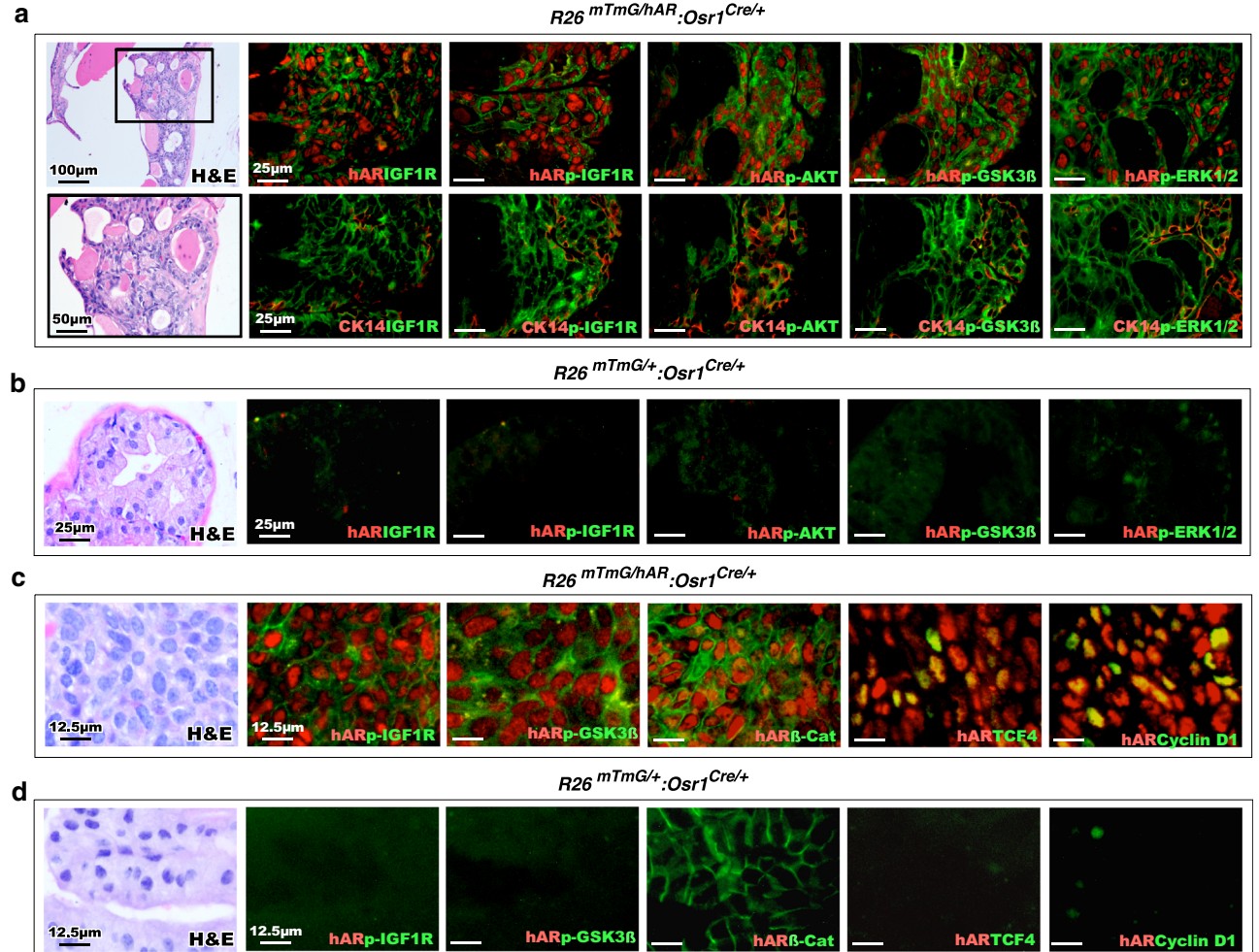

**Fig. 6 | Activation of hAR transgene expression initiates PIN formation and tumor development through aberrant elevation of IGF1 and ß-catenin signaling pathways.** Representative images of H&E and co-IF staining for hAR or CK14, a basal cell marker, and IGF1 signaling downstream mediators, including IGF1R, phosphorylated-IGF1R (p-IGF1R), p-AKT, p-GSK3ß, and p-ERK1/2 in PIN tissue sections from $R26^{mTmG/hAR}:Osr1^{Cre/+}$ mice at 6 months age (**a**) and normal prostate tissue sections from age-matched $R26^{mTmG/+}:Osr1^{Cre/+}$ wild type mice (**b**). Scale bars, 100 μm; 50 μm; 25 μm. Representative images of H&E and co-IF staining for hAR and IGF1 downstream (p-IGF1R, p- GSK3ß) or Wnt downstream (ß-catenin, *TCF4*, and *Cyclin D1*) in PCa tissue sections from $R26^{mTmG/hAR}:Osr1^{Cre/+}$ mice at 12-month age (**c**) and normal prostate tissue sections from age-matched $R26^{mTmG/+}:Osr1^{Cre/+}$ wild type mice (**d**). Scale bars, 12.5 μm. Representative images with consistent results from three independent experiments are shown.

activation of Wnt/β-catenin signaling pathways in prostate tumor cells of $R26^{mTmG/hAR}:Osr1^{Cre/+}$ mice, implicating the regulatory role of *hARtg* in activating Wnt/β-catenin to promote prostate tumor development and growth.

### Activation of human *AR* transgene expression initiates PIN formation and promotes tumor development through aberrant elevation of IGF1 and Wnt/β-catenin signaling pathways

It has been shown that aberrant activation of IGF1 signaling can either directly or via activating AKT and GSK3β axes stabilize cellular β-catenin to augment its activity for prostate tumor growth[28–31]. We then assessed the effect of IGF1 signaling in activating Wnt/β-catenin axes in *hARtg* + atypical basal cells. Co-IF analyses identified positive staining for IGF1R or phosphorylated IGF1R overlaid with transgenic AR staining in atypical HGPIN cells (Fig. 6a). Positive staining for phosphorylated forms of AKT, GSK3β, and ERK1/2 were also detected and overlaid with transgenic AR in those HGPIN cells on adjacent tissue sections (Fig. 6a). Positive staining for IGF1R, phosphorylated IGF1R, AKT, GSK3β, and ERK1/2 also overlaid with CK14 staining, demonstrating the basal cell

properties of those double positive atypical cells (Fig. 6a). In contrast, there is no or very weak staining with transgenic AR, IGF1R, and phosphorylated IGF1R, AKT, GSK3β, and ERK1/2 in prostate tissues of age- and sex-matched $R26^{mTmG/+}:Osr1^{Cre/+}$ controls (Fig. 6b). These data suggest up-regulated IGF1R expression and activation of IGF1R and its downstream target, AKT, leading to the phosphorylation of GSK3β, and ERK1/2 in atypical basal cells of HGPIN lesions. Using co-IF approaches, we further assessed the activation of IGF1R and Wnt/β-catenin axes in *hARtg*+ tumor cells. Co-staining for phosphorylated IGF1R and GSK3β, as well as nuclear β-catenin and its downstream targets, *TCF4* and *Cyclin D1*, with transgenic AR staining appeared specifically in prostate tumor cells (Fig. 6c) in comparison to normal prostatic epithelial cells of WT mouse prostate tissues (Fig. 6d), suggesting the regulatory loop of *hARtg* expression in elevating IGF1 signaling and activating Wnt/β-catenin axis during PIN and tumor development in $R26^{mTmG/hAR}:Osr1^{Cre/+}$ mice. These lines of scientific evidence are consistent with previous in vitro studies and provide more relevant data to demonstrate the co-regulation of androgen, IGF1, and Wnt/β-catenin axes in prostate tumor development and growth[28,29,31].

## Aberrant transgenic AR induces Wnt/β-catenin signaling activation to promote prostate cancer development

To gain dynamic and deep insight into the oncogenic role of *hARtg* in prostate tumorigenesis, we conducted single-cell trajectory analyses to unveil transcriptomic changes that were specifically induced by *hARtg* expression and governed cell fate decisions through PIN initiation and progression to prostatic adenocarcinomas[32]. Using Monocle2[32], we observed that the pseudotime trajectory plots of *hARtg*+ epithelial cells from merged PIN and PCa samples displayed a starting point composed mainly of BE2, a PIN cell cluster, that differentiated to the cell trend with luminal cell properties, and then further progressed to two main luminal cell branches mainly possessing LE1-2 tumor cells (Fig. 7a). In contrast, *hARtg*- epithelial cells showed a different trajectory fate, which started with BE1 cells and differentiated into a single luminal cell branch containing normal LE5-7 cells (Fig. 7b). The cellular properties of different cell branches were further assessed using trajectory expression plots with different cellular markers. The expression of *hARtg* in *mGFP*+ Osr1-lineage cells was only observed in *hARtg*+ cell samples (Fig. 7c). Whereas the expression of *Krt14* mainly appeared in the start of the branches, composing BE2 or BE1 cells, in *hARtg*+ or *hARtg*− cell plots, respectively, *Krt8* expression showed at later cell branches in pseudotime in both *hARtg*+ and *hARtg*−samples, representing their luminal cell properties (Fig. 7c, d). Intense expression of endogenous *Ar* and *Pbsn* appeared at the later trend of the luminal cell branch in *hARtg*- cells (Fig. 7d). In contrast, in *hARtg*+ cell plots, both endogenous *Ar* and *Pbsn* expression only appeared earlier in pseudotime and decreased as cells progressed toward the two luminal tumor cell branches, indicating their undifferentiated and transformed cellular properties (Fig. 7c). Importantly, the expression of Wnt/β-catenin target genes, including *Tcf4, Ccnd1, Axin2,* and *Lgr5,* was observed in later luminal cell branches of *hARtg*+ cells through cell trajectory plots in comparison to those of *hARtg*- cells (Fig. 7c vs d), implicating the activation of Wnt/β-catenin signaling during the course of tumor development. Using pseudotemporal kinetic analysis, we further demonstrated increased expression of *Tcf4, Ccnd1, Axin2,* and *Lgr5* in the later trend of tumor cells (fate 2) in comparison to the middle trend of cells (fate 1) in *hARtg*+ cell trajectory plots (Fig. 7e), showing the activation of Wnt/β-catenin signaling correlating with the course of tumor development. Taken together, the above analyses provide a dynamic and high-resolution image for transgenic AR induced Wnt signaling activation through prostate cancer development.

## Correlative activation of androgen, IGF1, and β-catenin signaling occurs in human prostate cancer samples

Next, we assessed aberrant AR, IGF1, and Wnt/β-catenin signaling pathways in both human primary PCa samples from the TCGA Pan-Cancer Atlas and advanced PCa samples in cBioPortal[33,34]. Aberrant activation of AR signaling through *AR* gene amplification or increasing its expression was detected in 5% of 488 primary PCa samples. Within these samples, about 42% and 55% of them also co-existed with aberrant alterations in *IGF1R* and Wnt/β-catenin downstream target genes, *MYC* and *CCND1*, respectively (Fig. 7f, top panel). Intriguingly, similar alterations in *AR* amplification and increasing expression were identified in 61% of 444 total advanced PCa samples. Increased co-occurrence of altered *IGF1R* and gain-of-function alterations and mutations in Wnt/β-catenin pathways also appeared (Fig. 7f, bottom panel). Odds ratio testing showed significant co-occurrences of alterations of *AR* and *IGF1R, MYC,* or *CCND1* genes and their transcripts in both primary and advanced PCa samples (Fig. 7g). Specifically, significantly positive correlations were observed between the transcripts of *AR* with *IGF1R, CTNNB1,* or *MYC* in primary PCa samples (Fig. 7h). Enrichment of AR on the *IGF1R* gene locus was also observed in AR ChIP-seq datasets of human PCa samples in comparison to controls with normal prostate tissues (Supplementary Fig. 6), further

supporting our mouse ChIP-seq analyses (Supplementary Fig. 5). These lines of scientific evidence provide the importance and clinical relevance of aberrant AR activation in altering IGF1 and Wnt/β-catenin signaling pathways in human prostate tumorigenesis.

## Aberrant activation of Wnt/β-catenin signaling pathways enhances AR-mediated prostate tumor growth

Multiple lines of evidence have shown a promotional role of Wnt/β-catenin in PCa growth and progression[35,36]. Specifically, an interaction between the AR and β-catenin has been identified in PCa cells, directly augmenting tumor cell growth[37–39]. Using organoid cultures derived from prostatic tumor cells of *R26^{mTmG/hAR}:Osr1^{Cre/+}* mice, we directly assessed the role of Wnt/β-catenin in *hARtg*+ tumor cell growth. The developed prostatic organoids were treated with the anti-androgen, enzalutamide (Enz), and Wnt inhibitors, ICG-001[40] and iCRT3[41] alone or in combination (Fig. 8a). It has been shown that iCRT3 can disrupt the interaction between AR and/β-catenin and inhibit AR-mediated transcription and cell growth in PCa cells[41]. Measuring average sizes of individual organoids and the organoid forming efficiency showed significant reduction in samples treated with Enz, ICG-001, and iCRT3 alone or in combination in comparison with vehicle-treated controls (Fig. 8b, c). Accordingly, more and larger organoids developed in vehicle-treated samples than in those treated with different inhibitors in the brightfield images (Fig. 8d). Specifically, far fewer and smaller organoids revealed in samples treated with Enz combined with Wnt inhibitors, ICG-001 or iCRT3. Histological analyses recapitulated similar prostatic adenocarcinoma lesions in vehicle-treated organoids as observed in tumor tissues of *R26^{mTmG/hAR}:Osr1^{Cre/+}* mice (Fig. 8d). Abnormal glandular structures resembling PIN lesions represented in samples treated with Enz, or Wnt inhibitors alone. Appearances of minor pathological changes to normal glandular structures were observed in samples treated with both Enz and Wnt inhibitors. IHC showed CK5 and CK8 staining in organoid cells, indicting their epithelial properties (Supplementary Fig. 7a1–b6). Positive nuclear staining for transgenic AR appeared in organoids treated with vehicle, ICG-001, and iCRT, whereas samples treated with Enz alone and in combination with Wnt inhibitors showed both nuclear and cytoplasmic staining for transgenic AR (Fig. 8d). Positive staining for *c-Myc* and *Cyclin D1,* the downstream targets of β-catenin, appeared in vehicle-treated samples and slightly in Enz-treated samples, but not in those treated with Wnt inhibitors alone or in combination with Enz (Fig. 8d). Measuring Ki67 positive cells in the organoid samples revealed the inhibitory effects with Enz and Wnt inhibitors. Among them, the most robust inhibition appeared in samples treated with Enz and iCRT3 (Fig. 8e). Taken together, these data demonstrate the inhibitory effects of both antiandrogen and Wnt inhibitors on the growth of *hARtg*+ tumor organoids.

Using in vivo tissue grafting assays, we further examined co-inhibition of AR and Wnt signaling in *hARtg*+ tumor growth. Specifically, we tested iCRT3[41], a Wnt inhibitor that also showed inhibition of AR signaling in PCa cells, alone or with Enz. Prostate tumor cells isolated from *R26^{mTmG/hAR}:Osr1^{Cre/+}* mice were implanted under the kidney capsule of SCID mice (Fig. 8g). Four weeks after implantation, SCID mice were administered Enz, iCRT3, or both, as well as vehicle, and analyzed four weeks post treatment. Although grafts treated with Enz or iCRT3 alone appeared significantly smaller in size and less in weight than vehicle-treated group, samples treated with both Enz and iCRT3 showed the lowest weights and smallest size among the groups (Fig. 8f). Histologically, vehicle-treated samples retained abnormal pathological characteristics of prostate adenocarcinomas as observed in prostate tumor tissues of *R26^{mTmG/hAR}:Osr1^{Cre/+}* mice whereas Enz or iCRT3-treated samples displayed minor pathological changes. Samples treated with both Enz and iCRT3 showed much less pathological changes than other controls (Fig. 8h). Positive staining for CK5 and CK8 was observed in all grafted samples (Supplementary Fig. 7g1–i4).

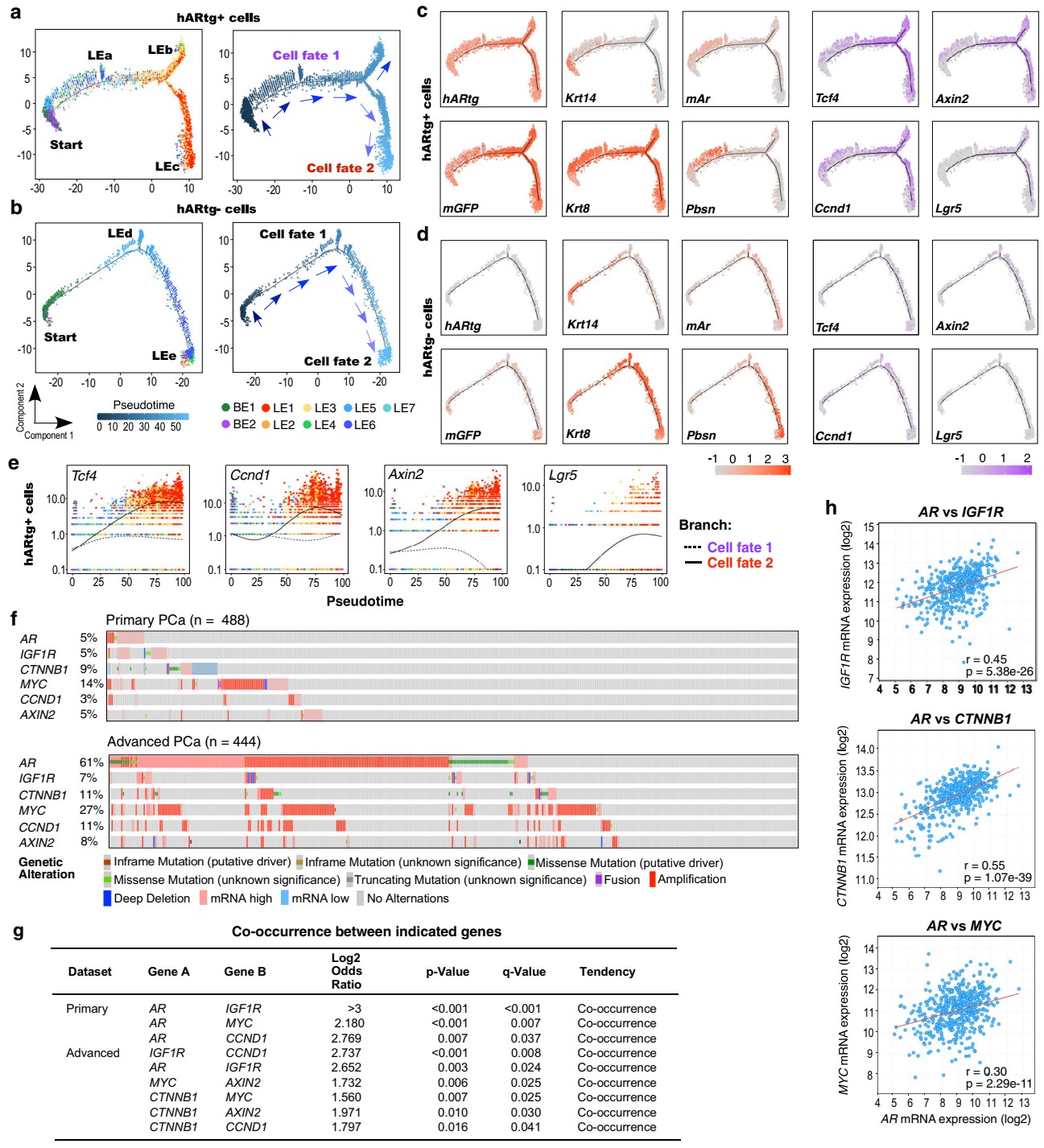

**Fig. 7 | Aberrant IGF1 and Wnt/β-catenin signaling pathways regulate prostate cancer development.** Pseudotime trajectory plots displaying a predicted directional path of *hARtg* + (**a**) or *hARtg*- epithelial cells (**b**) colored by cluster identity (left) and by pseudotime (right). Expression of the indicated genes projected onto the pseudotime trajectory in *hARtg* + (**c**) and *hARtg*- epithelial cells (**d**). Red and purple color intensities indicate the scaled expression levels of cellular markers, including *hARtg, mGFP, Krt14, Krt8*, endogenous *Ar* (*mAr*), *Pbsn*, and Wnt downstream target genes, including *Tcf4, Axin2, Ccnd1*, and *Lgr5*, respectively, in each cell. **e** Linear pseudotime expression plots for indicated Wnt downstream target genes in *hARtg* + epithelial cells. Lines on each plot correspond to the path of tumor progression moving from the start point (left) to indicated branch tips (right) as in **a** (right). **f** Oncoprint outlining genetic alterations and expression of the indicated

genes in 488 human primary prostate cancer samples and 444 metastatic castration-resistant prostate cancer samples. Datasets of Prostate Adenocarcinoma (TCGA, PanCancer Atlas) and Metastatic Prostate Adenocarcinoma (SU2C/PCF Dream Team) for primary and advanced prostate cancer, respectively, were extracted from cBioPortal. Colors show genetic alteration as indicated in legend. See also Methods section. **g** Mutual exclusivity panel analysis depicting the co-occurrence of alterations of the indicated genes in human primary PCa and metastatic castration-resistant prostate cancer. *P* values were computed using Fisher's exact test (one-sided) and adjusted using Benjamini-Hochberg correction. **h** Scatter plots showing the Spearman correlation of co-expression between *AR* and *IGF1R, CTNNB1*, or *MYC. r* = spearman's correlation coefficient, *p* = p value. Two-sided *t*-test. See also Supplementary Fig. 8.

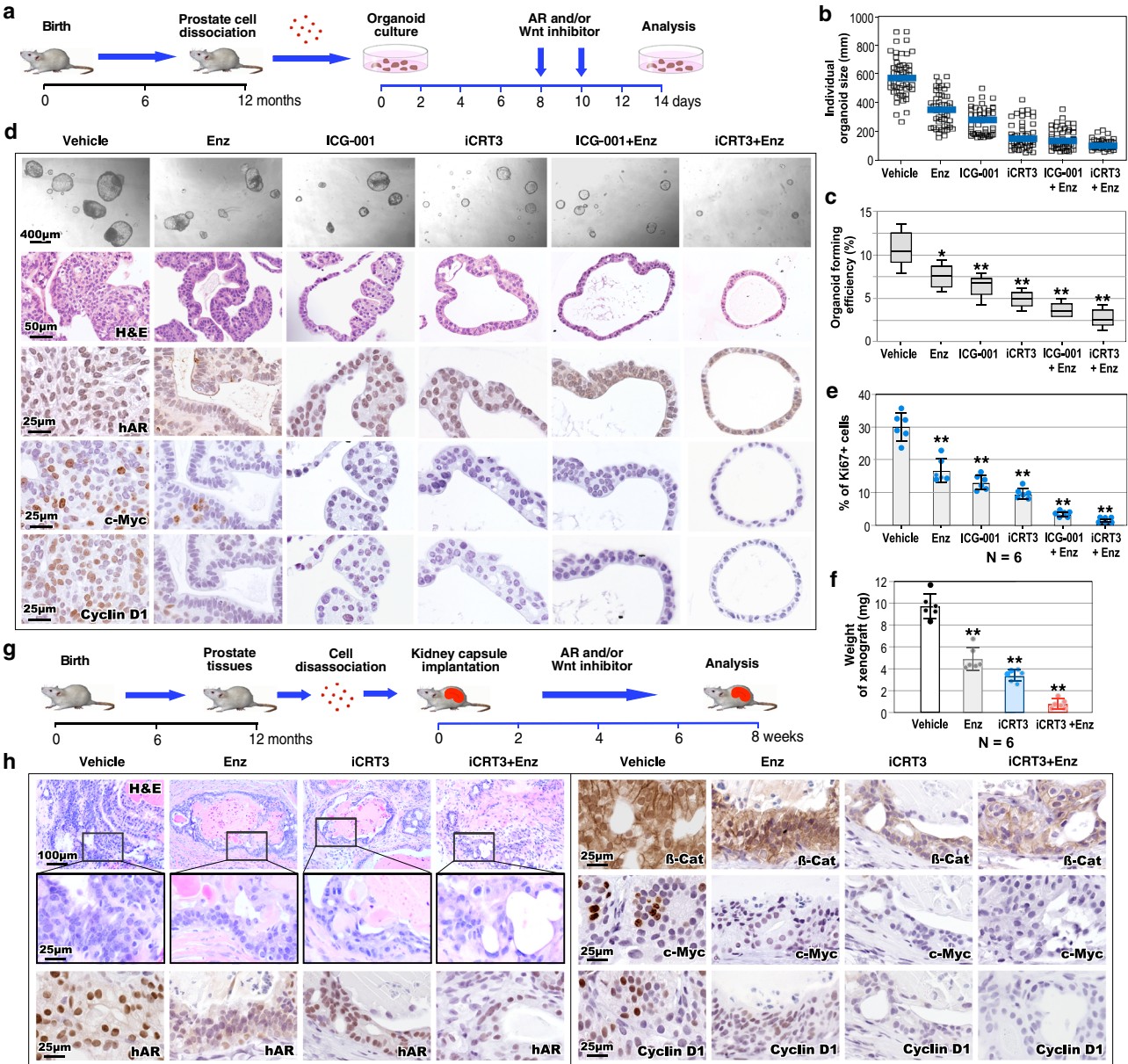

**Fig. 8 | Co-inhibition of androgen and Wnt/β-catenin signaling pathways in prostate tumor growth. a** Schematic representation of the experimental design for the ex vivo organoid culture performed. Organoids derived from PCa cells of *R26^(mTmG/hAR)*:*Osr1^(Cre/+)* mice were treated with vehicle, antiandrogen (10 μM Enzalutamide; Enz), Wnt inhibitor (10 μM ICG-001 or 10 μM iCRT3), or combination of Enz and ICG-001 or iCRT3 two times for 6 days. See also Methods section. **b** Quantification of individual organoid size. Organoids per treatment group (*n* = 50) examined over three independent experiments. The center blue bar indicates the median value in each group. **c** Quantification of organoid forming efficiency showing the percentage of organoids above 50 μm diameter per total cells seeded at day 0 in a well. The center line represents the median value; the box borders represent the lower and upper quartiles (25% and 75% percentiles, respectively); the ends of the bottom and top whiskers represent the minimum and maximum values, respectively. **d** Representative images of brightfield, H&E and IHC staining for the indicated antibodies of the organoids with the indicated treatments. Scale bars, 400 μm; 50 μm; 25 μm. **e** Quantification of Ki67 positive cells per total cells in the organoids with the indicated treatments. **f** Weights of xenografts (*n* = 6) from groups treated as indicated. Data are represented as mean ± SD (*n* = 6 replicates per data point). Student's *t*-test, **p < 0.01. **g** Schematic representation of the experimental design using in vivo kidney capsule transplantation. See also Methods section. **h** Histological and IHC analysis of graft tissues with the indicated treatment. Representative images of H&E and IHC images for indicated antibodies on adjacent sections from graft tissues with the indicated treatments. Scale bars, 100 μm; 25 μm. In **c**, **e**, and **f** data are represented as mean ± SD of six independent samples over three biological replicates. Two-sided *t*-test, *p < 0.05, **p < 0.01. Representative images with consistent results from three independent experiments are shown. Source data and the exact p-values are provided in the Source Data file.

While robust nuclear staining for hAR was observed in vehicle treated samples, much less staining appeared in the other treated samples (Fig. 8h). Positive nuclear staining for β-catenin revealed in the vehicle and Enz-treated grafts. In contrast, only cellular membrane staining of β-catenin appeared in samples treated with iCRT alone and in combination with Enz (Fig. 8h). Positive staining for *c-Myc* and Cyclin D1, the downstream targets of β-catenin, also only revealed in vehicle treated samples (Fig. 8h). These data reaffirm that the co-inhibition of AR and Wnt signaling pathways represses the growth of *hARtg*+ tumor cells.

## Discussion

Although the promotional role of androgen-signaling in prostate tumorigenesis has been implicated for many decades[2,6], the fundamental mechanisms by which the AR induces oncogenic

transformation and initiates PIN and prostate tumor development are still unclear. Conditional expression of human *AR* transgene in mouse prostatic *Osr1*-lineage cells results in HGPIN and prostatic adeno-carcinomas as mice progressed in age[10], recapitulating the oncogenic role of the AR as observed in human prostate cancer. Here, we directly evaluated the cellular properties and transcriptomic changes of prostatic *Osr1*-lineage cells through prostatic embryonic and pubertal development. The expression of *Osr1* revealed transiently in UGE tissues at E15.5 and E18.5 days, overlaying with other prostatic stem and progenitor cell markers. A robust increase in mGFP+ cells, the descendants of *Osr1*-expressing cells, was observed in UGE as well as prepubescent and pubertal prostatic epithelia. Genetically tracing labeled *Osr1*-expressing cells further showed their ability in expanding prostatic epithelia through embryonic to pubertal stages of prostate development. Identifying the basal epithelial cellular properties of mGFP+ cells in prepubescent prostatic epithelia further evidences *Osr1*-lineage cells functioning as prostatic progenitors. These observations are consistent with early studies showing the expression of *Osr1*-driven *Cre* starting at E11.5 in urogenital tissues that the prostate is derived from[42]. The expression of OSR1 was further identified in urogenital epithelial cells with the AR in human male embryonic tissues at 18 weeks (Supplementary Fig. 7j), implicating the biological importance and relevance of OSR1-lineage cells in human prostate development. These findings identify the cellular properties of prostatic *Osr1*-expressing cells and their descendants through prostate early, prepubescent, and pubertal development, elucidating a cell population of prostate epithelial progenitors.

Development of HGPIN and prostate adenocarcinomas in *R26^{mTmG/hAR}:Osr1^{Cre/+}* mice but not in *R26^{mTmG/+}:PB^{Cre/+}* counterparts, aligning with the identification of *hARtg*+ atypical and tumor cells specifically within those HGPIN and prostate adenocarcinomas lesions, demonstrates the direct role of transgenic AR expression in inducing PIN and PCa development[10]. Data from our scRNA-seq analyses further identified elevated IGF1 and insulin related signaling pathways in *hARtg*+ basal epithelial cells in *R26^{mTmG/hAR}:Osr1^{Cre/+}* mice, suggesting a regulatory mechanism underlying aberrant AR action in stimulating IGF1 signaling to promote PCa initiation. Early studies have shown that individuals with high circulating levels of IGF1 have an increased risk of PCa, particularly advanced disease[43,44]. Activated IGF1R through a ligand-receptor interaction can promote mitogenic signaling events through increasing cell proliferation and inhibiting apoptosis[45]. Aberrant IGF1R and insulin receptor expression has been frequently observed in human PCa samples[46]. Additionally, over-expression of the *Igf1* in mouse prostatic basal epithelia induced oncogenic transformation and mouse PIN development[21]. Moreover, the AR has been shown to directly regulate the transcription of the *IGF1R* in PCa cells[24]. Our current findings are consistent with the previous results, demonstrating a critical role of *hARtg* in elevating aberrant IGF1 axis in *Osr1*-lineage basal cells to induce prostate oncogenic transformation and PIN formation. Specifically, our ChIPseq and ChIP-qPCR data revealed an enrichment of transgenic AR in the regulatory regions of the mouse *Igf1r* gene locus in the *hARtg*+ cells. These lines of evidence implicate a regulatory mechanism underlying aberrant AR activation through IGF1-mediated signaling pathways in PCa development. These findings also suggest an unknown interaction between AR and IGF1 signaling pathways that manipulate PCa metabolism and contribute to the pathogenesis of PCa.

In *R26^{mTmG/hAR}:Osr1^{Cre/+}* mice, HGPIN lesions enable to continue to progress to prostatic adenocarcinomas, resembling what occurs in human PCa. This intriguing pathologic characteristic has not been observed in previous AR transgenic mice using the PB promoter[7,47,48]. In this study, we identified aberrant Wnt/β-catenin signaling activation in *hARtg*+ prostatic tumor cells, uncovering the regulatory mechanisms underlying AR action in atypical cells to progress to prostatic

adenocarcinomas. The activation of aberrant Wnt/β-catenin axis has been shown to directly promote prostate cancer cell proliferation as well as tumor development and progression[30,49]. β-catenin is a co-regulator of the AR and synergistically promotes prostate tumor cell growth through direct interaction with the AR[50]. Moreover, IGF1 signaling can enhance the stability of cellular β-catenin and augment the AR-mediated transcription in PCa cells[28,29]. In this study, we provided multiple lines of scientific evidence demonstrating a regulatory loop through aberrant activation of androgen signaling in stimulating IGF1 and β-catenin activation to promote PCa initiation and development. The above data also suggest that the current AR transgenic model is a biologically significant and clinically relevant tool for the future investigation of AR action in prostate tumorigenesis.

Using single-cell trajectory analyses, we evaluated the effect of *hARtg* expression on cell fates and differentiation during the course of PIN initiation and progression to prostatic adenocarcinomas. In *hARtg*+ cell pseudotime trajectory plots, BE2, a PIN cell cluster, appeared at the starting branch, gradually progressing towards cells that possess endogenous *Ar* and *Pbsn* expression and are mainly composed of LE3-4 luminal cells, and finally progressed to two luminal tumor cell branches comprising LE1-2 clusters. In contrast, *hARtg*- cells showed a very different cell trajectory trend, which started with BE1 cell cluster and progressed to the cell trend and branch with normal luminal cell properties. Using Slingshot tools[51], we further validated the above results and observed similar trajectory fates of *hARtg*+ and *hARtg*- cells as observed in our Monocle2 analyses (Supplementary Fig. 8a, b). These data provide a dynamic image for the different cell fates initiated from *hARtg*+ or *hARtg*- cells. Selective up-regulation of Wnt/β-catenin signaling downstream target gene expression, such as *Tcf4*, *Ccnd1*, *Axin2*, and *Lgr5*, only revealed in *hARtg*+ luminal tumor cell branches in comparison to both the earlier trend of *hARtg*+ cells (cell fate 1) in pseudotime and the cell branch of *hARtg*− cells in trajectory expression plots, demonstrating the activation of Wnt/β-catenin signaling induced by hAR aligning with PCa development and progression. Using newly developed package, PseudotimeDE, we further addressed uncertainty of pseudotime DEG inference, validating the expression of *hARtg* and Wnt/β-catenin downstream target genes along the established trajectory branches[52] (Supplementary Fig. 8c).

In this study, we also identified the significant correlation in co-occurrence of altered AR, IGF1R, and Wnt/β-catenin signaling pathways in both primary and advanced human prostate cancer samples. Notably, these alterations and mutations appeared significantly higher in the advanced tumor samples than in primary tumor samples. To gain direct evidence on the regulatory role of AR action in IGF1R and β-catenin activation, we specifically analyzed the expression of AR, IGF1R, and β-catenin using human primary PCa samples. We observed the significant correlation between the expression of the *AR* with *IGF1R*, *CTNNB1*, or *MYC* gene. Analyses of human ChIP-seq datasets also showed increased enrichment of AR on the *IGF1R* gene locus in human PCa samples in comparison to normal prostate tissues. These data are consistent with our observations in *R26^{mTmG/hAR}:Osr1^{Cre/+}* mice, and further support the regulatory role of aberrant AR activation in initiating prostatic tumorigenesis and promoting tumor growth and progression through elevating IGF1 and Wnt/β-catenin signaling pathways.

Using both antiandrogens and Wnt inhibitors, we evaluated the effect of co-targeting androgen and Wnt/β-catenin signaling pathways on the growth of *hARtg*+ prostatic tumors in organoid cultures and in-vivo tissue grafting models. Whereas the antiandrogen, Enz, and Wnt inhibitors, ICG-001 or iCRT3 all showed the inhibitory effect on the growth of organoids derived from *hARtg*+ tumor cells, iCRT3 revealed more potent effect than Enz or ICG-001. Additionally, iCRT3 in combination with Enz showed the most inhibitory effect among all different treated groups. In tissue grafting assays, the similarly robust inhibition was also observed in prostate tumor grafts treated with

iCRT3 and Enz combined. It has been shown that iCRT3, a Wnt inhibitor, can also disrupt the interaction between AR and β-catenin and repress AR-mediated transcription and growth in prostate cancer cells[41]. These data provide relevant and important evidence for developing future therapeutic strategies to co-target AR and Wnt/β-catenin signaling pathways for treating advanced prostate cancer. They also support this AR transgenic model being a biologically significant and clinically relevant tool for future investigation.

## Methods

### Ethics statement

All experimental procedures and care of animals in this study were carried out according to the Institutional Animal Care and Use Committee (IACUC) at Beckman Research Institute at City of Hope (California, US) and approved by the IACUC. Euthanasia was performed by $CO_2$ inhalation followed by cervical dislocation.

### Mouse generation, mating and genotyping

*Rosa26mTmG/+ (R26mTmG/+)* reporter[53], *Osr1-Cre*[42], and *Rosa26hAR^LoxP/wt*, also named *R26hAR/+* mice[10] were obtained and used in this study, respectively. The *PB^Cre/+* mice were obtained from the NCI mouse repository (strain #: 01XF5)[54]. *R26mTmG/hAR*, *R26mTmG/+:Osr1^Cre/+*, or *R26mTmG/+:Pb^Cre/+* mice were first produced and then used to generate *R26mTmG/+:Osr1^Cre/+* and *R26mTmG/hAR:Osr1^Cre/+* mice. Experimental mice generated in this study were mixed from C57BL/6 and 129S1/SvImJ backgrounds. Mice were genotyped after collecting mouse tail tips by PCR approaches using specific primers (Supplementary Table 2). All mice bearing tumors were closely monitored during the entire course of the study according to the guidelines of IACUC in our institution, and the maximal tumor size and burden in mice was never exceeded in this study.

### Isolation of normal, PIN, and PCa cells from mice

Cells were isolated and dissected from the mouse UGS or prostate tissues at the indicated timepoint[55]. Once trimmed, prostate tissues were cut into small pieces and digested in Gibco™ Dulbecco's Modified Eagle's Medium (DMEM)/Ham's F-12 50/50 Mix (DMEM/F12) (11320033, Fisher Scientific) supplemented with 10% Gibco™ Fetal Bovine Serum, 1 nM dihydrotestosterone (DHT) (A8380, Sigma Aldrich), 10 μM Y-27632 dihydrochloride (M1817, Abmole Bioscience), and 1 mg/ml type II collagenase (17101-015, Life Technologies) at 37 °C for 135 min with rotation at 150 rpm, and then in Gibco™ TrypLE (12605-028, Fisher Scientific) supplemented with 1 nM DHT, 10 μM Y-27632 dihydrochloride and 0.5 U/μl DNase I (D5025, Sigma Aldrich) at 37 °C for 30 min with rotation. Cells were filtered through a 37 μm cell strainer (27215, StemCell technologies) and resuspended in PBS with 0.05% Bovine Serum Albumin. Cell viability and number were detected using a TC Automated Cell Counter (Bio-Rad Laboratories), and only cells at least 80% viability were processed.

### Histological, IHC and IF analyses

Mouse tissues were cryo-protected in 30% sucrose in phosphate-buffered saline (PBS) and then embedded in OCT (Tissue–Tek®, Sakura Finetek) or processed in paraffin following fixation in 10% neutral-buffered formalin (American Master Tech Scientific). Membrane-bound Tomato (mT) and membrane-bound green fluorescent protein (mGFP) signal was detected by mTmG assays[56]. Briefly, OCT tissue blocks were cut to 5 μm and sections were washed with PBS, pH 7.3, and then were mounted with Vectashield Mounting Medium with DAPI (H-1200, Vector Laboratories). 4 μm serial sections from paraffin-embedded tissues were used for histological analyses using Hematoxylin-eosin (H&E) staining, immunohistochemistry (IHC) staining, or immunofluorescence (IF) staining[27]. Pathological analyses were conducted in accordance with the guidelines suggested by The Mouse Models of Human Cancers Consortium Prostate Pathology Committee in 2013[7]. For IHC staining, tissue slides were boiled in

antigen retrieval buffer, 0.01 M citrate buffer (pH 6.0) or Tris/EDTA (pH 9.0). Following rehydration, slides were placed in 0.3% $H_2O_2$ in methanol for 15 min, blocked in 5% goat serum for 1 hr, and incubated with primary antibodies in 1% goat serum in PBS overnight at 4 °C. Tissues were then incubated with biotinylated secondary antibodies for 1 hr and with horseradish peroxidase (HRP)-conjugated streptavidin (SA-5004, Vector Laboratories) followed by visualization using DAB kit (SK-4100, Vector Laboratories). Sections were counterstained with hematoxylin and mounted with Permount Mounting Medium (SP15–500, Fisher Scientific). For IF staining, tissues sections underwent heat-induced epitope retrieval, blocked in 5% normal goat serum, and incubated with primary antibodies at 4 °C overnight and with fluorescent-conjugated secondary antibodies, and then mounted with Vectashield Mounting Medium with DAPI (H-1200-10, Vector Laboratories). All antibodies used in this study are detailed in Supplementary Table 3.

### Microscope image acquisition

Images of H&E and IHC staining were taken by an Axio Lab A1 microscope using ×10, ×20, and ×40 Zeiss A-Plan objectives and were captured using a Canon EOS 1000D camera and AxioVision software (Carl Zeiss). Images of mTmG signals and IF staining were acquired on a Nikon ECLIPSE E800 epi-fluorescence microscope using ×10, ×20, and ×40 Nikon Plan Fluor objectives using an QImaging RETGA EXi camera and QCapture software (QImaging).

### Single-cell RNA-seq analyses

Upon isolation, cells were loaded on the Chromium Controller (10X Genomics) targeting 7000–10,000 cells per line. Single cell RNA-seq libraries were prepared using the Chromium v2 single cell 3′ RNA-seq reagent kit (10X Genomics) according to the manufacturer's protocol. The purity and size of cDNA were validated by capillary electrophoresis on Agilent 2100 Bioanalyzer using Agilent High Sensitivity DNA Kit (#5067-4626, Agilent Technologies). Libraries were sequenced on a NovaSeq 6000 instrument (Illumina) with a depth of 50K–100 K reads per cell. Processing of raw sequencing data, including generation of FASTQ files and counting of Unique Molecular Identifiers (UMI), was conducted using the 10x Genomics Cell Ranger pipeline (3.1.0). Reads were aligned to the mm10 genome with *mGFP* and *hARtg* sequences[15] for gene expression count. The Seurat package (3.2.1) in R (3.6.3) was used for the subsequent data analysis following upload of a filtered feature bar coded matrix including 1031, 5899, 8983, 7225, and 11,084 cells from E18.5 male UGS, P14, P35, PIN, and PCa prostate tissues, respectively. For a quality-control step, potential empty droplets and multiplets were filtered out based on dead cells or low-quality cells with a fraction of mitochondrial RNA higher than 10–15% were eliminated (Supplementary Figs. 1 and 4). Following this final filtering, 6332 PIN cells with an average of 3751 genes per cell and 21297 UMI counts per cell, and 9412 PCa cells with an average of 3532 genes per cell and 19,899 UMI counts per cell were conserved for future analyses. Integrating datasets were performed on Seurat using *FindIntegrationAncors* and *IntegrateData* functions with twenty dimensions. Normalized and scaled data were clustered by Graph-based clustering approach using Seurat functions including *RunPCA* with 30 principal components, *FindNeighbors* with twenty dimensions, and *FindClusters* with 0.5 resolutions. Data was visualized by a nonlinear dimensionality reduction UMAP technique using the Seurat *RunUMAP* function.

For analysis of signaling pathways, differential expression analysis was conducted on Seurat using *FindMarkers* or *FindAllMarkers* functions. *p* values were adjusted for multiple testing with Benjamini-Hochberg correction. Genes with |average Log2 fold-change| > 1 and adjusted *p* value < 0.05 were considered as differentially expressed genes (DEGs). We used canonical pathway analysis in Ingenuity Pathway Analysis (IPA) (v01-20-04) and Gene Set Enrichment Analysis (GSEA) (4.1.0) analysis with hallmark and curated gene sets. Spearman

pairwise correlation matrices as the measure of association between *hARtg* and other genes were analyzed and plotted using the *ggcorr* function from the GGally package R (https://www.rdocumentation.org/packages/GGally/versions/1.5.0/topics/ggcorr) in R.

For pseudotime generation and trajectory analysis, Seurat objects were converted into CellDataSet format and the DDRTree algorithm[32] applied to reduce expression profiles into two dimensions using the Monocle2 package (2.12.0)[32] in R. Using BEs as the starting point, Monocle reconstructed differentiation trajectories and defined pseudotime. Branch expression analysis modeling (BEAM) was used to detect genes significantly branching between cell fate 1 and cell fate 2 and to identify genes that change after fate decisions along the *hARtg* in PCa development. Next, we generated aligned kinetic curves for Wnt target genes, which showed significant difference in the BEAM test, using Monocle's *plot_genes_branched_pseudotime* function. Pseudotime trajectories were also identified using the Bioconductor package Slingshot (1.8.0)[51] by default, and visualized on a 2-dimensional UMAP embedding. For pseudotime-based differential expression testing, Pseudotime DE package (1.0.0)[52] was used with SingleCellExperiment objects, processed with Slingshot[51], as input for the pseudotime inference method. After subsampling by default, differentially expressed gene testing was performed using the *run PsuedotimeDE* function with Negative Binomial Model. The gene trajectories estimated by the fitted Generalized Additive Model were visualized.

## RNA isolation and RT-qPCR

Total RNA was extracted from fresh mouse prostate tissues or organoids derived from prostate tissues of *R26^{mTmG/+}:Osr1^{Cre/+}* or *R26^{mTmG/hAR}:Osr1^{Cre/+}* mice using RNeasy Mini Kit (#74104, Qiagen) and reverse-transcribed using SuperScript IV First-Strand Synthesis System (#18091050, Fisher Scientific) according to the manufacturer's protocol. RT-qPCR were performed on the 7500 Real-Time PCR system (Fisher Scientific) using Power-SYBR Green PCR Master Mix (4367659, Applied Biosystems) with specific primers (Supplementary Table 3). Relative quantification was normalized to the level of peptidylprolyl isomerase A (Ppia) and was calculated using the comparative $C_T$ method[57].

## ChIP-seq and ChIP-qPCR

ChIP-DNA was obtained through ChIP assay[58]. Briefly, mouse prostate tissues were minced, cross-linked with 1% formaldehyde for 25 min at room temperature (RT) and quenched with 150 mM glycine for 10 min. Following wash with cold PBS, samples were homogenized in cell lysis buffer (140 mM NaCl, 50 mM Tris-HCl [pH 8.0], 1 mM EDTA, 10% Glycerol, 0.5% NP-40, and 0.25% Triton X-100). The chromatin was resuspended in nuclear lysis buffer (0.2% SDS, 10 mM Tris-HCl [pH 8.0], 1 mM EDTA, and 0.5 mM EGTA) and fragmented to an average size of 200–500 bp with a Sonic Dismembrator Model 100 (Fisher Scientific) at 4 °C. After centrifugation, the cell lysate was diluted with ChIP dilution buffer (0.01% SDS, 167 mM NaCl, 16.7 mM Tris-HCl [pH 8.1], 1.1% Triton X-100, and 1.2 mM EDTA), pre-cleared using Dynabead Protein G (10003D, Invitrogen) and then was subjected to immunoprecipitation with Dynabead Protein G conjugated with AR antibody (sc-7305X, Santa Cruz Biotechnology) with rotation for 4 hr at 4 °C. Beads were washed sequentially with washing buffer 1 (0.1% SDS, 150 mM NaCl, 20 mM Tris-HCl [pH 8.0], 1% Triton X-100, and 2 mM EDTA), washing buffer 2 (0.1% SDS, 500 mM NaCl, 20 mM Tris-HCl [pH 8.0], 1% Triton X-100, and 2 mM EDTA), washing buffer 3 (250 mM LiCl, 10 mM Tris-HCl [pH 8.0], 1% NP-40, 1% deoxycholate, and 1 mM EDTA), TE buffer (50 mM Tris-HCl [pH 8.0] and 10 mM EDTA). DNA was eluted with elution buffer (1% SDS, 50 mM Tris-HCl [pH 8.0] and 10 mM EDTA), incubated with 0.2 M NaCl and 1 mg/mL Protease K at 65 °C for 6 hr, purified with phenol/chloroform/isoamylalcohol (Invitrogen) and was precipitated with glycogen and cold EtOH. ChIP DNA samples were subjected to amplification using MicroPlex Library Preparation kit

(AB-004-0012, Diagenode) on a StepOnePlus Real-Time PCR System (Applied Biosystems). The libraries were quantified by Qubit dsDNA HS Assay Kit and validated with Agilent Bioanlyzer DNA high Sensitivity Kit. Following library generation, DNA was purified twice with Agencourt AMPure XP beads (A63880, Beckman Coulter) and sequenced on Illumina Hiseq 2500 Sequencer. For ChIP-seq data analysis, reads were mapped to mouse genome assembly mm9 using Novoalign v3.02.07. MACS v2.0[59] was used to call Peaks using the corresponding negative control from *R26^{mTmG/+}:Osr1^{Cre/+}* and to generate bed graph files for visualization of peaks. Gene annotations of mm9 genome were downloaded from the RefSeq database. Binding profiles were visualized using Integrated Genome Viewer (IGV v.6.3)[60]. ChIP-qPCR was performed with specific primers (Supplementary Table 2) using Power-SYBR Green PCR Master Mix (Applied Biosystems) on the 7500 Real-Time PCR Detection System (Fisher Scientific).

## RNAseq and data analysis

RNA integrity and quality were evaluated using Agilent RNA 6000 Nano Kit (#5067-1511, Agilent Technologies). cDNA libraries were prepared using Kapa RNA HyperPrep Kit with RiboErase (KR1351, Kapa Biosystems) followed by sequencing on Illumina Hiseq 2500 at the City of Hope Integrative Genomics Core. Adapter sequences and other redundant sequences were removed from raw paired-end reads using cutadapt (v1.9.1)[61]. The filtered reads were aligned to the mm9 mouse reference genome using HISAT2 (2.1.0)[62]. Read counts per gene were detected using HTSeq (v0.6.1)[63] and were further normalized using the trimmed mean of *M*-value method[64] or the upper-quartile method[65] in edgeR (v2.26.7)[66] to obtain counts per million-mapped reads (CPM) and transcripts per million (TPM). Only genes with a CPM > 1 in at least two samples were retained for differential analysis. The fold change, *p*-value and FDR were calculated using quasi-likelihood (QL) *F*-test in edgeR settings with default using normalized expression values, TPM. The *p* values were adjusted for multiple testing with Benjamini-Hochberg correction. DEGs were defined as those having a |Log2 fold-change| > 1 and adjusted *p*-value < 0.05 and were preranked based on the Log2 fold-change and adjusted *p*-value. Clipper tools was also used with logCPM as input at the target FDR threshold $q = 10\%$, for controlling false positive data[67]. GSEA (4.1.0) using preranked gene lists were performed for annotation, integration, and visualization of database in order to identify network and biological functions of DEGs[68]. Concordant results between two normalization methods were considered significant and results from the trimmed mean of *M*-value method normalization were presented as representative data.

## Human dataset analysis

An integrative analysis of AR with IGF1/Wnt-related genes and clinical characteristics was conducted using cBio Cancer Genomics Portal (https://cbioportal.org), which is an open-access resource for the interactive analysis and visualization of multidimensional cancer genomics datasets[34]. The gene alteration frequency, co-occurrence, and co-expression were analyzed using prostate cancer datasets from Prostate Adenocarcinoma (TCGA, PanCancer Atlas; 488 samples) (https://www.cell.com/pb-assets/consortium/pancanceratlas/pancani3/index.html) for primay PCa and Metastatic Prostate Adenocarcinoma (SU2C/PCF Dream Team, 444 samples) for advanced PCa[33]. Co-expression analysis was performed using a log-rank test to identify the significance of the Spearman's correlation coefficient between the mRNA expression z-scores (RNASeq V2 RSEM). We obtained AR ChIPseq datasets from both human primary PCa tissues, including "GSE130408", "GSE114737", and "GSE56288", and normal prostate tissues (GSE130408) that were extracted from the Gene Expression Omnibus (GEO)[69]. Those data were visualized with ENCODE[70] and RefSeq tracks on UCSC genome browser[71]. Slides from paraffin-embedded tissues of human UGS at 18 weeks were used for IHC analyses[72].

## Organoid culture and kidney capsule implantation

Prostatic tumor tissue was isolated from $R26^{mTmG/hAR}:Osr1^{Cre/+}$ mice at 12 months of age and dissociated into single cells by digestion in DMEM/F12/FBS/Collagenase for 2 hr and then TrypLE for 30 min at 37 °C. Digested cells were passed through 37 μm cell strainers and resuspended in 1:1 (v/v) PBS:Matrigel (#356231, BD Biosciences). For organoid culture, approximately 2000 cells per well were seeded in 24 well-plate and cultured in DMEM/F12 containing 0.25 μM A83-01 (R&D Systems), 1× B27 (Life Technologies), 1 nM DHT, 10 ng/mL EGF (PeproTech), 100 ng/mL N-acetylcysteine (PeproTech), 100 ng/mL Noggin (PeproTech), 2.5 ng/mL R-spondin1 (R&D Systems) and 100 μM Y-27632, as reported in the previous study[55] but reduced concentration for EGF and R-spondin1. Eight days after culture, organoids were treated with vehicle, dimethyl sulfoxide (DMSO), antiandrogen, 10 μM Enzalutamide (Enz), Wnt inhibitors, 10 μM ICG-001 (HY-14428, Med-ChemExpress), 10 μM iCRT3 (HY-103705, MedChemExpress), or combination of Enz and ICG-001 or iCRT3 two times for 6 days and were fixed in 10% neutral-buffered formalin. Fixed cells were subjected to Histogel (HS-4000-012, Fisher Scientific) and paraffin embedding for histological analysis. Individual organoid size was quantified with the Image J (NIH) using at least 100 organoids per group. Organoid forming efficiency (%) was determined by quantifying the percentage of organoid structure above 50 μm diameter per total cells seeded at day 0 in a well. Quantification of the percentage of Ki67+ cells per total cells were performed from five different areas in each sample. All experiments were replicated with two different mice in triplicate wells.

For in vivo kidney capsule implantation, approximately $1 \times 10^5$ PCa cells were transplanted under the renal capsule of 6 to 8-week-old male NOD/SCID mice. 4 weeks after implantation, mice were randomized into vehicle control and treatment groups and were administered with vehicle (10% DMSO in corn oil), oral Enz at 20 mg/kg body weight, intraperitoneal iCRT3 at 30 mg/kg body weight[41,73,74], or combination of Enz and iCRT3. The compound was administered three times per week for 28 days. Xenografts were excised, weighed and fixed for histological analysis.

## Quantification and statistical analysis

Data are presented as the mean values ± SD for the indicated number of independently performed experiments. All data are representative of the results of at least three independent experiments. The significance of the differences between data ($^*p < 0.05$, $^{**}p < 0.01$) was measured using 2-tailed Student's $t$-test. Adjusted $p$ values were corrected for multiple testing using the Benjamini–Hochberg's procedure. Differentially expressed gene (DEG) lists were determined using a Wilcoxon Rank Sum test, with genes showing adjusted $p$-value < 0.05 defined to be significant. Spearman's correlation coefficient > 0.3 and $p$ value < 0.05 was considered to indicate a statistical significance. Enrichment scores (ES) for each gene set in the ranked list of genes were calculated by a running-sum statistic using GSEA[68]. Nominal $p$ values of ES were estimated using an empirical phenotype-based permutation test and corrected for multiple hypothesis testing using FDR. As recommended by the GSEA User Guide, pathways with FDR < 0.25 were considered significant in exploratory GSEA pathway analysis.

## Reporting summary

Further information on research design is available in the Nature Research Reporting Summary linked to this article.

## Data availability

Raw data of RNA-seq, single cell RNA-seq, ChIP-seq have been deposited in Gene Expression Omnibus database under accession number GSE164971. Source Data for this study are provided with this paper. All relevant data in this study are available within the article, Supplementary information, or Source Data. Source data are provided with this paper.

## Code availability

The bioinformatics analyses were conducted using open-source software, including Cell Ranger version 3.1.0, Seurat version 3.2.1[11], R version 3.6.3, Novoalign version 3.02.07, MACS version 2.0[59], IGV version 6.3[60], cutadapt version 1.9.1[61], HISAT2 version 2.1.0[62], HTSeq version 0.6.1[63], edgeR version 2.26.7[66], GSEA version 4.1.0[68], and IPA version 01-20-04. R scripts used to process sequencing data are available in "GitHub repository [https://github.com/wk-kim/hAR-OSR1_Prostate_Tumorigenesis]"[75].

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

## Acknowledgements

This work was supported by NIH grants R01CA070297, R01CA166894, R01DK104941, and R01CA233664. The authors wish to thank Dr. Gerald Cunha and Dr. Laurence Baskin for their help of our IHC experiments.

## Author contributions

Z.S. and W.K.K. conceived the project and designed the experiments. W.K.K., J.M., D.H.L., C.H.N., A.J.B, and Z.S. generated mouse colonies and performed genotyping and tissue collection experiments. W.K.K., A.W.O., J.M., D.H.L., V.L., A.H., J.A., and Z.S. performed related mouse works and collected RNA, DNA and single cell samples for the analyses. J.W., S.Y., W.K.K., and A.W.O. performed sequence experiments and analyzed sequencing data. W.K.K., V.L., D.H.L., A.H., and R.C. conducted ex vivo and in vivo experiments and performed staining and analyses. All authors analyzed and confirmed data. W.K.K, A.W.O., and Z.S. wrote the manuscript.

## Competing interests

The authors declare no competing interests.
