## [Peer Review File · Nature Communications]

Aberrant androgen action in prostatic progenitor cells induces oncogenesis and tumor development through IGF1 and Wnt axesReviewers' comments:

Reviewer #1 (Remarks to the Author):

In this manuscript, the authors used single-cell transcriptomic and genetic tracing analyses to uncover the underlying mechanisms driven by AR to increase IGF1-signaling pathway in Osr1-expressing and lineage cells to initiate PIN formation. They found that accumulation of Wnt/b-catenin accumulates in atypical PIN cells promotes tumor development and inhibition of Wnt represses the growth of hARTg+ prostate tumor cells in ex-vivo and xenograft. Overall, the work seems to be quite shallow of its actual significant findings as well as the statistical analysis and it is heavily based on association which represents a major limitation of the work. Greater focus on the importance of Osr1 could add a novel aspect to the work.

Major comments:

- Cells having the same phenotype but in different phases of the cell cycle often form quite distinct clusters. The authors should consider the effects of the cell cycle during their analysis other than to label a single cluster as "proliferating cells" in the single cell analysis. They should show UMAP plots with cells coloured by cell-cycle phase and by ensuring that any results (e.g. DEGs) from the scRNA-seq analyses are not just consequences of different clusters having different numbers of cells in each cell-cycle phase.
- Figure 1, Osr1 expression is claimed to be "significantly" less in P14 and P35 compared to UGS, but some quantification however statistical tests should be performed. E.g. number of cells expressing both Osr1 and a particular prostatic stem/progenitor cell marker above some threshold could be stated, and/or the gene-gene correlations between Osr1 and Prom1, Itga, Ly6a, Tacstd2, Trp63 could be computed. Some of the markers in Fig 1E (e.g. Ly6a) do not appear to have much coexpression with Osr1.
- What about co-expression of Osr1 with two or more of the stem/progenitor markers? From Fig 1E, it appears that different stem/progenitor markers are coexpressed with Osr1 in different populations of cells (they are not all co-expressed by the same cells) – what is the significance of this?
- p5 L87: Sentence is not grammatically correct. Do the authors mean e.g. "were revealed", "appeared", or "were present" instead of "revealed"? The same grammatical error is made on p7 L142, p9 L183, p12 L244, p13 L273, p15 L315, p15 L323, and p16 L347.
- P7 L121-123: This is a strong overstatement of the data described prior. The authors have not demonstrated a "driver role" of hARTg expression. They have shown an association between hARTg expression in Osr1-lineage cells and development of PIN and PCa, but they cannot actually conclude from the data described that hARTg is a driver. In other words, the authors have established correlation but not causality.
- p9 L167-170: This is a very simple approach to investigate the regulatory role of hARTg in basal epithelial (BE) cells that does not make full use of the information available from single-cell data. The authors should present gene-gene Spearman correlation values for the union of the BE1 and BE2 clusters. This would more directly show correlations between hARTg expression and other genes in BE cells and be more appropriate for inferring regulatory relationships between genes to address the technical artifact (See e.g. <https://doi.org/10.1016/j.patter.2021.100211>). The authors could consider the use SCENIC (<https://scenic.aertslab.org/>).
- Fig 3D and Fig 4C: Please show the positions of the medians and the lower and upper quartiles on the violin plots. Also, please show the adjusted p values (e.g. q values) from the differential gene expression analysis for the genes shown in the violin plots.
- In multiple places, the authors state numbers of DEGs having p values < 0.05. Were these p values adjusted for multiple hypothesis testing (e.g. using the Benjamini-Hochberg procedure)? If not, then they need to be, and this should be stated in the text.
- I like Fig 5C, but the significance of the enrichment (i.e. q value) needs to be shown alongside the enrichment scores.
- p13 L274: "differential potentials to luminal cells" does not make grammatical sense.
- p12 L255 to p14 L284 and Fig 6A-E: This entire section is fundamentally flawed and should either be replaced or removed. Pseudotime trajectories are based on distances between data points (cells) in state space and can only be accurately inferred where the data has sufficient coverage of all cell phenotypes along the true biological trajectory of cell differentiation. The authors should have recognized that the presented pseudotime analysis does not make biological

sense because it suggests that hARtg- normal luminal cells were derived from hARtg+ basal cells, which is not well supported. Otherwise, the analysis should be discarded and no conclusions should be drawn from it.

- Fig 6F-J seems to be totally distinct from Fig 6A-E so should not have belonged to the same figure. In fact, if Fig 6A-E is completely removed as I suggest above, then Fig 6F-J would actually follow on better from the previous section (corresponding to Fig 5) than Fig 6 currently does. However, although Fig 6F-J is very nice, it is very weak without substantial additional work. For a start, the same H&E and IF experiments need to be performed in normal prostate tissue. Without showing that hAR-negative normal prostate cells do not have the same levels of staining of p-IGF1R, p-AKT, p-GSK3beta, p-ERK1/2, beta-catenin, Tcf4 and Cyclin D1 as hAR+ PIN/PCa cells, Fig F-J is largely meaningless. See also next comment related to Fig 6F-J.
- p14 L294-296 and L298-300: The authors massively overstate the conclusions that can be drawn from the data in Fig 6F-J. Without the normal-prostate control IF experiments, no conclusions can be made on the basis of Fig 6F-J. Even with the normal-prostate control, the authors still would not be able to make such strong conclusions. At best, they would have shown correlations between expression of hAR, activation of IGF1R and AKT, and expression of beta-catenin and its downstream targets in PIN and PCa, but they would not have established causality.
- p16 L334-335: "in an organoid model" should be added to this sentence, since the authors have not demonstrated that Wnt/beta-catenin signalling drives prostate tumour growth in vivo.
- Fig 7B,E: The figure or figure legend needs to state the specific inhibitors that were used. In Fig 7E, it needs to be indicated that the inhibitor (iCRT3) was a co-inhibitor for the AR and Wnt pathways and not just a Wnt inhibitor, if this is indeed true (this is what L335-336 of the main text implies). If iCRT3 only inhibits AR signalling indirectly through its effects on beta-catenin, then this needs to be stated in the text.
- Fig 7E-H: If iCRT3 inhibits AR signalling through any mechanism other than its effects on Wnt/beta-catenin (which is what the authors imply by describing it as a co-inhibitor of the AR and Wnt signalling pathways), then AR and Wnt each need to be inhibited independently in order to deconvolve the effects of inhibiting AR and Wnt on tumour growth. Otherwise, the decreased xenograph weight in iCRT3-treated mice compared to vehicle-treated mice could mostly be due to the effects of iCRT3 on AR. In any case, the effects of iCRT3 should be compared to a specific inhibitor of AR signalling.
- Consideration of the clinical relevance of the work presented in this manuscript is limited to a single figure panel (Fig 7A) and a single paragraph in the text. This is not sufficient to support the claim. For example, it would be easy to look at correlations between the expression levels of the various relevant genes in public data from human cohorts. Gene expression correlations in human patients would also be more relevant to this manuscript than co-occurrences of genomic alterations are.
- p26 L567-568: RPKM is not a suitable normalization method for differential analysis of RNA-seq data. Trimmed mean of M-values (L564) is acceptable. Other normalization methods may also be suitable (e.g. upper quartile, TPM, etc.). The best approach to RNA-seq analysis is to perform the analysis using multiple different normalization methods and identify concordant results between the different methods, since differential expression analysis can be strongly influenced by the choice of normalization.

Minor comments:

- p3 L35: I would clarify that ADT eventually fails in most patients. Current wording suggests that ADT is mostly useless.
- Fig 3C legend needs more detail. What is the "ratio" that is plotted in the figure panel?
- Some panels need to be enlarged

Reviewer #2 (Remarks to the Author):

Previous studies have been focused on pathways of insulin-like growth factor and androgen receptor and vice versa in prostate cancer. These are two important oncogenic pathways in prostate. The experiments in the present study are innovative and the models interesting. Most results are clearly presented. There are some problems because it is not always clear whether AR signaling in this paper is characterized as "wild-type" or "altered" signaling. Several suggestions for the improvement follow:

1. Page 15, a total of 488 primary PCa samples are mentioned. However, it is not clear whether these samples are derived from patients who received any kind of therapy or not. This is very important for interpretation of data because androgenic response (or androgen ablation) may affect regulation of the IGF pathway.
2. Is amplification the only change which occurs for androgen receptor?
3. It is not clear whether activation of the Wnt pathway may have the same effect as activation of the AR pathway because these two pathways may have different activities in terms of regulation of cellular stemness. Please clarify.
4. The relevance of the paper may be improved if data on Osr-1 expression in human material are included.
5. Fig. 2O, are individual clusters shown in that Figure associated with IGF signaling?
6. Gi. 7 will be more meaningful if stem cell marker expression is included.

Reviewer #3 (Remarks to the Author):

In this manuscript, using a murine prostate cancer model R26mTmG/hAR:Osr1Cre/+ and single cell RNA-seq technique, the author tried to explore the mechanism of AR-driving prostate cancer. They identified that Igf1 and Wnt signaling pathway were upregulated at high-grade prostatic intraepithelial neoplasia (HGPIN) and invasive prostatic adenocarcinoma (PCa) lesions in mice. They claimed that Igf1 and Wnt signaling were the downstream targets of AR. Finally, to treat the murine PCa, they administrated the organoids which derived from PCa cells of R26mTmG/hAR:Osr1Cre/+ with WNT inhibitor. Overall, the concept, animal model and therapeutic strategy in the study are not novel, although the single cell RNA-seq data from mice are beautiful and solid. Many concepts have been reported. I also didn't see many human relevant data in the manuscript.

It seems this murine prostate cancer model (R26mTmG/hAR:Osr1Cre/+) is more relevant to human PCa compared with Pten-knock out murine PCa model. Could the author comment this? Is R26mTmG/hAR:Osr1Cre/+ mice PCa model resistant to castration/AR antagonist? If so, the author could focus on its mechanism. If not, the author could establish a castration/AR antagonist resistant model using R26mTmG/hAR:Osr1Cre/+ model. Another question is that I don't understand why the author select WNT inhibitor to treat the mice PCa in this study. According to the results from this study, Igf1 and Wnt signaling are the AR downstream targets. If so, treatment with AR antagonist may be enough to inhibit these signaling pathways and the progression of PCa. WNT signaling pathway is broad in the body, so the treatment with WNT inhibitors may have more side effects than AR inhibitors.

1. For the single cell RNA-seq experiments in Figure 3, it seems the control samples are not quite suitable. For example, the authors tried to explore the impact of AR overexpression in Osr1+ prostatic progenitor. In this case, the experimental group should be R26mTmG/hAR:Osr1Cre/+ and control group should be R26mTmG/+ :Osr1Cre/+, which just have one variate (hAR). Then we can easily compare the gene expression in the mGFP positive cells between these two groups. However, they just separated hARTg-negative and hARTg-positive basal epithelial cells. These two types of basal epithelial cells may originally express different genes because they belong to Osr-negative and Osr-positive basal epithelium. The analysis in Figure 3 could mislead the study.
2. Another question is how many mice for one group were used in the single cell RNA-seq? This is important because the difference between individuals may also impact the results.
3. I am very curious that Osr1-Cre-driven hARTg but not PB-Cre-driven hARTg transgenic mice can develop prostatic adenocarcinoma lesions. Why does PCa specifically occur in Osr1+ prostatic progenitor? What is the role of Osr1 during the tumorigenesis? Does AR overexpression reprogram the transcriptome and epigenome in Osr1+ prostatic progenitor at early stage? On the other hand, Osr1 not only expresses in prostatic progenitor but also expresses in other cells such as fibroblasts and smooth muscle cells (Fig. 1B-D), indicating the specific epigenetic or cellular environment in Osr1+ prostatic progenitor may be important for the oncogenesis function of AR. In other words, the other factors but not just AR and Osr1 are also important for the progression of prostatic adenocarcinoma. The authors indicated that AR upregulated Igf1 and Wnt signaling in prostatic progenitor, but why overexpression of AR didn't impact the same pathways in other Osr1+ cells?
4. Although the author showed that AR can directly bind to the chromatin at Igf1 loci, we still don't know if hAR can directly regulate the transcriptional level of Igf1. AR has about 60,000 binding sites at chromatin, but AR does not regulate the transcriptional level for every gene which has AR

binding sites. To confirm if AR regulates the transcriptional level of Igf1 or Wnt signaling, more experiments such as treatment the organoid with AR agonist/antagonist (DHT/enzalutamide) and qRT-PCR should be done. The same experiments in mice should be confirmed in human prostate cancer organoids or cell lines.

5. In Figure 1H2 and I2, Osr1 is a transcript factor, but why does most of Osr1 locate at membrane? Could the author comment this?

6. Does Osr1 specifically expressed in prostate prostatic epithelial cell? If not, we may see hAR expression in other type of cells such as fibroblasts and smooth muscle cells. Could the author comment this?

7. "TP63+mGFP+ basal epithelial cells were only observed in prostate tissues of R26mTmG/+;Osr1Cre/+ mice but not in those of R26mTmG/+;PBCre/+ mice (Supplementary Fig.S3G1-1' versus S3G2-2')". What is the role of Osr1+ basal epithelial cells? Does Osr1+ basal epithelial cells mainly drive oncogenesis and tumor development in this model?

8. In the abstract, the author mentioned human prostate cancer, but I didn't see many human relevant data in this manuscript. Does AR overexpression upregulate IGF1 and WNT signaling in human prostate cancer?

Reviewer #1 (Remarks to the Author):

In this manuscript, the authors used single-cell transcriptomic and genetic tracing analyses to uncover the underlying mechanisms driven by AR to increase IGF1-signaling pathway in Osr1-expressing and lineage cells to initiate PIN formation. They found that accumulation of Wnt/b-catenin accumulates in atypical PIN cells promotes tumor development and inhibition of Wnt represses the growth of hARtg+ prostate tumor cells in ex-vivo and xenograft. Overall, the work seems to be quite shallow of its actual significant findings as well as the statistical analysis and it is heavily based on association which represents a major limitation of the work. Greater focus on the importance of Osr1 could add a novel aspect to the work.

We greatly appreciate the Reviewer's comments and have taken them to heart and directly addressed them with new experimental results and analyses. In this study, we directly addressed an important and long-term outstanding question regarding the fundamental mechanisms underlying AR functioning as a tumor promoter in inducing prostatic oncogenesis. The early study has shown that conditional expression of human AR transgene in *Osr1*-lineage cells induces both high-grade PIN (HGPN) and prostatic tumor lesions in murine prostates, which recapitulates AR oncogenic action and mimics what occurs in human prostate cancer. Using this biologically relevant AR transgenic mouse model, single cell RNA sequencing analyses, and other *ex-vivo* and xenograft models, we identified that transgenic AR expression in prostatic *Osr1*-lineage cells induces IGF1-axis activation and HGPN development and that aberrant IGF1-signaling further activates Wnt/ β -catenin pathways to promote atypical PIN progression to prostate tumors. These findings demonstrate a novel mechanism underlying oncogenic AR signaling in prostate tumorigenesis. Additionally, this study is also very different from most previous studies using probasin (PB) promoter-driven prostate cancer mouse models. In fact, although the PB promoter has been frequently used in the past, it is an AR downstream promoter, postnatally activated in prostatic epithelium, only expressed in mouse tissues, and has significant limitation for studying ligand-independent tumor growth and progression. Given these significant challenges in the field, in this current study, we mainly focused on characterizing the pathogenesis of prostate tumor developed in the AR *Osr1-Cre* transgenic mice rather than the expression of *Osr1* itself. Below, we specifically addressed each of the Reviewer's comments.

Major comments:

- *Cells having the same phenotype but in different phases of the cell cycle often form quite distinct clusters. The authors should consider the effects of the cell cycle during their analysis other than to label a single cluster as "proliferating cells" in the single cell analysis. They should show UMAP plots with cells coloured by cell-cycle phase and by ensuring that any results (e.g. DEGs) from the scRNA-seq analyses are not just consequences of different clusters having different numbers of cells in each cell-cycle phase.*

We greatly appreciate the Reviewer's point, and have modified the related cell clusters (see the current Fig. 1b) and also provided UMAP plots presenting different "cell-cycle phases" in the current revised manuscript (see Supplementary Fig. 2a).

- *Figure 1, Osr1 expression is claimed to be "significantly" less in P14 and P35 compared to UGS, but some quantification however statistical tests should be performed. E.g. number of cells expressing both Osr1 and a particular prostatic stem/progenitor cell marker above some threshold could be stated, and/or the gene-gene correlations between Osr1 and Prom1, Itga, Ly6a, Tacstd2, Trp63 could be computed. Some of the markers in Fig 1E (e.g. Ly6a) do not appear to have much coexpression with Osr1.*

We truly appreciate the expert point of the Reviewer. In this revision, we provided 1) the percentage of cells that express *Osr1* and other stem cell marker in total prostatic cells and epithelial cells at different

time points (Supplementary Fig. 2d and 2e), and 2) the gene-gene Spearman correlation values between *Osr1* and other cellular markers in both total epithelial and stromal cells (Fig. 1d and Supplementary Fig. 1f). Interestingly, we only observed a significant correlation between the expression of *Osr1* with several stem/progenitor cell markers in total epithelial cells but not in stromal cells at E18.5 samples, implicating the significance of *Osr1*-expressing cells in prostatic epithelial development.

• *What about co-expression of Osr1 with two or more of the stem/progenitor markers? From Fig 1E, it appears that different stem/progenitor markers are co-expressed with Osr1 in different populations of cells (they are not all co-expressed by the same cells) – what is the significance of this?*

As mentioned above, we have modified the above figure based on our new analyses following the Reviewer's suggestion. The revised plots provide additional evidence demonstrating co-expression between *Osr1* and other stem/progenitor markers in prostatic epithelial cells.

• *p5 L87: Sentence is not grammatically correct. Do the authors mean e.g. “were revealed”, “appeared”, or “were present” instead of “revealed”? The same grammatical error is made on p7 L142, p9 L183, p12 L244, p13 L273, p15 L315, p15 L323, and p16 L347.*

We apologize for those grammar errors and have corrected them in the current revision.

• *P7 L121-123: This is a strong overstatement of the data described prior. The authors have not demonstrated a “driver role” of hARTg expression. They have shown an association between hARTg expression in Osr1-lineage cells and development of PIN and PCa, but they cannot actually conclude from the data described that hARTg is a driver. In other words, the authors have established correlation but not causality.*

As reported in the previous study (Zhu et al, *J. Biol. Chem.* 286, 33478-88, 2011) and this manuscript, only $R26^{hAR/+};Osr1^{Cre/+}$ mice but $R26^{hAR/+}$ and $Osr1^{Cre/+}$ control littermates developed HGPI and prostate tumor lesions, indicating that transgenic AR expression regulated by *Osr1-Cre* directly contributes to the above phenotypes. In response to the Reviewer, we modified the sentence as follows: “The expression of both hARTg and mGFP in atypical and tumor cells within HGPI and prostate adenocarcinoma lesions demonstrate the critical role of transgenic AR in *Osr1*-lineage cells in inducing prostatic oncogenesis and promoting PIN and PCa development.” (see the end of Page 7).

• *p9 L167-170: This is a very simple approach to investigate the regulatory role of hARTg in basal epithelial (BE) cells that does not make full use of the information available from single-cell data. The authors should present gene-gene Spearman correlation values for the union of the BE1 and BE2 clusters. This would more directly show correlations between hARTg expression and other genes in BE cells and be more appropriate for inferring regulatory relationships between genes to address the technical artifact (See e.g. <https://doi.org/10.1016/j.patter.2021.100211>). The authors could consider the use SCENIC (<https://scenic.aertslab.org/>).*

We appreciate the expert opinion of the Reviewer. In response to the Reviewer, we presented additional data using gene-gene Spearman correlation analyses in the current revision (see Fig.3e). The current data are consistent with our findings and support the correlation between the expression of *hARTg*, *Igf1r* and its downstream target genes. Additionally, we also performed SCENIC analyses as the Reviewer suggested during the analyses, and observed similar regulatory changes as identified in our current analyses. Because our current mouse model contains the human *AR* transgene, we carefully identified the molecular changes induced by hARTg in *Osr1*-lineage BE and other epithelial cells using different analytic tools.

• *Fig 3D and Fig 4C: Please show the positions of the medians and the lower and upper quartiles on the violin plots. Also, please show the adjusted p values (e.g. q values) from the differential gene expression analysis for the genes shown in the violin plots.*

In response to the Reviewer, we provided the medians and the lower and upper quartiles as well as the adjusted p values as box plots in the revised Fig. 3d and Fig. 4c.

• *In multiple places, the authors state numbers of DEGs having p values < 0.05. Were these p values adjusted for multiple hypothesis testing (e.g. using the Benjamini-Hochberg procedure)? If not, then they need to be, and this should be stated in the text.*

In our original analyses, we identified DEGs with p values adjusted by the Bonferroni procedure using FindMarkers or FindAllMarkers function in R. In response to the Reviewer, we re-adjusted p values using the Benjamini-Hochberg procedure for the IPA and GSEA, and provided the data from these analyses with newly adjusted p values (see Fig. 3c, Fig. 4f, Fig. 5c, Fig.5d and Methods). We also stated the related changes in the “Methods” section and related figure legends.

• *I like Fig 5C, but the significance of the enrichment (i.e. q value) needs to be shown alongside the enrichment scores.*

In response to the Reviewer, we have provided q value for the above analyses and modified Fig. 5c in the current revision.

• *p13 L274: “differential potentials to luminal cells” does not make grammatical sense.*

In response to the reviewer (please also see our response below), we deleted this sentence in the current revision.

• *p12 L255 to p14 L284 and Fig 6A-E: This entire section is fundamentally flawed and should either be replaced or removed. Pseudotime trajectories are based on distances between data points (cells) in state space and can only be accurately inferred where the data has sufficient coverage of all cell phenotypes along the true biological trajectory of cell differentiation. The authors should have recognized that the presented pseudotime analysis does not make biological sense because it suggests that hARTg- normal luminal cells were derived from hARTg+ basal cells, which is not well supported. Otherwise, the analysis should be discarded and no conclusions should be drawn from it.*

We have carefully considered the Reviewer’s point. To gain deep understanding on the regulatory role of hAR during the course of PIN development and tumor progression, we re-performed single-cell trajectory analyses using hARTg+ cells from PIN and tumor samples (Fig. 7a-e). We observed a coordinated cell fate and differentiation trend from BE2, a PIN cell cluster, to LE1-2, tumor cell clusters, that comprises hARTg+ cells merged from both PIN and PCa samples. In contrast, hARTg- cells showed a very different differentiation trend. Importantly, the up-regulation of Wnt/ β -catenin signaling has been observed specifically in hARTg+ tumor cell branches, mainly composed of LE1-2 cells, providing new, in-depth, and dynamic images for Wnt/ β -catenin activation in prostate tumor development. Additionally, BE2, rather than BE1 cluster, was localized at the start of the hARTg+ cell trajectory plot, which is opposite of the hARTg- cell plot. These data provide additional evidence demonstrating that BE2 cells comprise of hARTg+ basal cells, possess PIN initiation cell properties, and directly contribute to PIN and tumor formation. We fully appreciate and share the Reviewer’s caution regarding the pseudotime trajectory analysis and realize it is still in the developing phases. Thus, we have carefully verified the current results with our other analyses and have only presented here the data that are also supported by other experimental approaches and analyses.

• *Fig 6F-J seems to be totally distinct from Fig 6A-E so should not have belonged to the same figure. In fact, if Fig 6A-E is completely removed as I suggest above, then Fig 6F-J would actually follow on better from the previous section (corresponding to Fig 5) than Fig 6 currently does. However, although Fig 6F-J is very nice, it is very weak without substantial additional work. For a start, the same H&E and IF experiments need to be performed in normal prostate tissue. Without showing that hAR-negative normal prostate cells do not have the same levels of staining of p-IGF1R, p-AKT, p-GSK3beta, p-*

ERK1/2, beta-catenin, Tcf4 and Cyclin D1 as hAR+ PIN/PCa cells, Fig F-J is largely meaningless. See also next comment related to Fig 6F-J.

We appreciate the Reviewer's comment. In the current revision, we provided the new panel of staining data that include normal prostate tissues from $R26^{mTmG/+};Osr1^{Cre/+}$ control mice to directly address the regulatory role of hAR expression in prostate cancer cells (see Fig. 6).

• *p14 L294-296 and L298-300: The authors massively overstate the conclusions that can be drawn from the data in Fig 6F-J. Without the normal-prostate control IF experiments, no conclusions can be made on the basis of Fig 6F-J. Even with the normal-prostate control, the authors still would not be able to make such strong conclusions. At best, they would have shown correlations between expression of hAR, activation of IGF1R and AKT, and expression of beta-catenin and its downstream targets in PIN and PCa, but they would not have established causality.*

We have carefully considered the Reviewer's critical comment. As indicated above, we have modified the above figures and provided controls from normal prostate tissues as the Reviewer suggested. In addition, multiple lines of previous evidence have shown the regulatory role of IGF1R in stabilizing and enhancing β -catenin activity through activated AKT and its downstream GSK3 β (Playford et al, PNAS 97:12103-8, 2000; Verras et al, Mol Endocrinol 19:391-98, 2005). Therefore, as detailed in this study, data from this Figure in combining with results from sc-RNA-seq, RNA-seq, qRT-PCR and other analyses are consistent with the earlier findings and implicate the underlying mechanisms for aberrant AR activation in regulating β -catenin prostate tumor cells. In response to the Reviewer, we have modified the related text and referenced the earlier studies to appropriately address the above points (see the beginning of page 15).

• *p16 L334-335: "in an organoid model" should be added to this sentence, since the authors have not demonstrated that Wnt/beta-catenin signalling drives prostate tumour growth in vivo.*

Based on our updated results, we modified the related section in this revision (see the end of page 18).

• *Fig 7B,E: The figure or figure legend needs to state the specific inhibitors that were used. In Fig 7E, it needs to be indicated that the inhibitor (iCRT3) was a co-inhibitor for the AR and Wnt pathways and not just a Wnt inhibitor, if this is indeed true (this is what L335-336 of the main text implies). If iCRT3 only inhibits AR signalling indirectly through its effects on beta-catenin, then this needs to be stated in the text.*

In this study, we used iCRT3, a Wnt inhibitor. The results from previous study also showed that iCRT3 can also disrupt the interaction between AR and β -catenin to reduce AR-mediated transcription and cell growth in prostate cancer cells (Lee et al, PNAS 110:15710-15, 2013). In response to the reviewer, we have modified the related text and provided the related information in the current revision (see the end of page 17).

• *Fig 7E-H: If iCRT3 inhibits AR signalling through any mechanism other than its effects on Wnt/beta-catenin (which is what the authors imply by describing it as a co-inhibitor of the AR and Wnt signalling pathways), then AR and Wnt each need to be inhibited independently in order to deconvolve the effects of inhibiting AR and Wnt on tumour growth. Otherwise, the decreased xenograph weight in iCRT3-treated mice compared to vehicle-treated mice could mostly be due to the effects of iCRT3 on AR. In any case, the effects of iCRT3 should be compared to a specific inhibitor of AR signalling.*

In response to the Reviewer's comment, we provided additional experimental data to show the effects of anti-androgen, enzalutamide (Enz), and Wnt inhibitor, ICG-001, and iCRT3 alone or in combination on the growth and development of organoids derived from hARtg+ tumor cells of $R26^{mTmG/hAR};Osr1^{Cre/+}$ mice (Fig. 8a-i). Additionally, we also examined the inhibitory effects of Enz and iCRT3 alone or in combination using *in vivo* tissue grafting assays (Fig. 8j-p). The current data provide clear scientific

evidence demonstrating more potent inhibitory effect of Enz and Wnt inhibitors than each reagent alone.

Consideration of the clinical relevance of the work presented in this manuscript is limited to a single figure panel (Fig 7A) and a single paragraph in the text. This is not sufficient to support the claim. For example, it would be easy to look at correlations between the expression levels of the various relevant genes in public data from human cohorts. Gene expression correlations in human patients would also be more relevant to this manuscript than co-occurrences of genomic alterations are.

As the Reviewer suggested, we provided more clinical evidence to demonstrate the interactions and regulatory loops between AR, IGF1, and Wnt signaling pathways in human prostate tumorigenesis. First, we identified the co-occurrence of alterations and mutations of *AR* and *IGF1R*, *CTNNB1*, or *MYC* genes and their expression in both primary and advanced tumor samples (Fig. 7f). Additionally, a significant correlation between those abnormalities was further identified in the above samples (Fig. 7g). Moreover, we demonstrated a significant positive correlation between the transcripts of *AR* with *IGF1R*, *CTNNB1*, or *MYC* in primary PCa samples (Fig. 7h). Furthermore, we showed an enrichment of AR on the *IGF1R* gene locus in ChIP-seq datasets of human PCa samples in comparison to controls with normal prostate tissues (Supplementary Fig. 6). These data further support the significance and relevance of aberrant activation of AR, IGF1, and Wnt/ β -catenin signaling pathways in human prostate tumorigenesis.

• p26 L567-568: RPKM is not a suitable normalization method for differential analysis of RNA-seq data. Trimmed mean of M-values (L564) is acceptable. Other normalization methods may also be suitable (e.g. upper quartile, TPM, etc.). The best approach to RNA-seq analysis is to perform the analysis using multiple different normalization methods and identify concordant results between the different methods, since differential expression analysis can be strongly influenced by the choice of normalization.

We really appreciate the Reviewer's expert suggestion. We originally normalized read counts using the trimmed mean of M-value method and calculated normalized expression values using TPM in edge R (see the "Methods"). In response to the Reviewer, we re-analyzed our RNA-seq data using the upper quartile normalization methods (see Supplementary Tables 5-7). We observed similar enrichments as identified in our previous analyses. We accordingly presented the concordant results from these two normalization methods in the current revision. We have stated the above changes in the current "Methods" section.

Minor comments:

• p3 L35: I would clarify that ADT eventually fails in most patients. Current wording suggests that ADT is mostly useless.

As the Reviewer suggested, we made the according change in the above sentence (see the end of the first paragraph on page 3).

• Fig 3C legend needs more detail. What is the "ratio" that is plotted in the figure panel?

In response to the Reviewer, we provided information for the "Ratio" in the revised legend, meaning "the number of significantly expressed genes compared with the total number of genes associated with the canonical pathway".

Some panels need to be enlarged

In the current revision, we have gone through each figure and made necessary adjustment.

Reviewer #2 (Remarks to the Author):

Previous studies have been focused on pathways of insulin-like growth factor and androgen receptor and vice versa in prostate cancer. These are two important oncogenic pathways in prostate. The

experiments in the present study are innovative and the models interesting. Most results are clearly presented. There are some problems because it is not always clear whether AR signaling in this paper is characterized as "wild-type" or "altered" signaling. Several suggestions for the improvement follow:

We greatly appreciate the Reviewer's comments regarding the significance and novelty of this study. In fact, although both IGF1 and androgen signaling pathways have been implicated in prostate tumorigenesis, the underlying mechanisms for these pathways to interact with each other to promote prostate cancer initiation and progression is largely unknown. One of the significant challenges is the lack of biologically relevant *in vivo* models to recapitulate the oncogenic role of AR signaling in prostate tumorigenesis. In this study, using new and relevant AR transgenic mice, single-cell RNA sequencing, and other experimental approaches, we demonstrated a regulatory role of AR in activating IGF1 signaling, which further cumulates Wnt/ β -catenin activation, in prostatic *Osr1*-lineage cells to initiate prostate oncogenesis and tumor development. These data directly address one of the most important questions in the field regarding how the AR functions as a tumor promoter to initiate prostatic oncogenesis and induce prostate cancer development. We also carefully went through each of the Reviewer's points and addressed them accordingly below.

1. Page 15, a total of 488 primary PCa samples are mentioned. However, it is not clear whether these samples are derived from patients who received any kind of therapy or not. This is very important for interpretation of data because androgenic response (or androgen ablation) may affect regulation of the IGF pathway.

We greatly appreciate the reviewer's point and have revised the figure as the reviewer suggested. Those primary samples are extracted from TCGA datasets (PanCancer Atlas). Based on the provided information, these are surgical biopsy specimens collected from patients that had not received prior treatment (<https://gdc.cancer.gov/about-data/publications/pancanatlas>, and Saltz et al., *Cell Reports*, 23, 181-193, 2018).

2. Is amplification the only change which occurs for androgen receptor?

Many studies have shown different abnormalities in androgen signaling pathways during prostate cancer development and progression (Please see Fig. 7f). Specifically, the *AR* gene amplification has been observed in one third of prostate cancer after ADT (Koivisto et al., *Cancer Res.*, 57, 314-9, 1997; Ruizeveld de Winter et al., *Am. J. Pathol.*, 144, 735-46, 1994), underscoring the importance of AR in disease progression.

3. It is not clear whether activation of the Wnt pathway may have the same effect as activation of the AR pathway because these two pathways may have different activities in terms of regulation of cellular stemness. Please clarify.

We appreciate the Reviewer's insightful point. Emerging evidence has shown that Wnt signaling pathways play a significant role in prostate development and tumorigenesis. Specifically, increased nuclear β -catenin expression has been shown to directly promote prostate tumor cell proliferation (Cheshire et al., *Oncogene*, 21, 8453-69, 2002). Conditional expression of stabilized β -catenin in prostate epithelium induces the development of PIN and prostate cancer (Bierie et al., *Oncogene*, 22, 3875-87, 2003; Gounari et al., *Oncogene*, 21, 4099-107, 2002; Lee et al., *Oncogene*, 35, 702-14, 2016). A protein-protein interaction between the AR and β -catenin has also been identified in prostate cancer cells (Mulholland et al., *J. Biol. Chem.*, 277, 17933-43, 2002; Truica et al., *Cancer Res.*, 60, 4709-13, 2000; Yang et al., *J. Biol. Chem.*, 277, 11336-44, 2002). Aberrant activation of Wnt signaling pathways has been frequently observed in advanced human prostate cancer (Robinson et al., *Cell*, 161, 1215-28, 2015). In this study, we demonstrate that aberrant AR activation elevates IGF1 signaling and subsequently stimulates Wnt activation in prostate *Osr1*-lineage cells through prostate oncogenesis and PIN and prostate tumor development. Data from organoids and *in vivo* tissue grafting assays further

showed that co-targeting AR and Wnt signaling produces more potent inhibition on prostate tumor growth than each reagent alone, further implicating the interaction between these two signaling pathways in prostate cancer growth. In response to the review, we have made several modifications in the current revision to emphasize the above scientific points.

4. The relevance of the paper may be improved if data on Osr-1 expression in human material are included.

We greatly appreciate the critical comment made by the Reviewer. We have provided IHC results showing Osr1 expression in human embryonic urogenital tissues (please see Supplementary Fig. 7j).

5. Fig. 2O, are individual clusters shown in that Figure associated with IGF signaling?

In Figure 2O, we showed the propositional cell counts for each epithelial cluster cells. As detailed in the manuscript, we identified an enrichment of IGF1 signaling in BE2 cluster that contains most basal hARtg+ epithelial cells. Additionally, using scRNA-seq and other experimental approaches, we identified a regulatory role of hARtg in inducing IGF1 signaling in BE2 basal epithelial cells. In response to the Reviewer, we emphasized our findings in the current revision (see the end of the first paragraph on page 9).

6. FiG. 7 will be more meaningful if stem cell marker expression is included.

In response to the Reviewer, we provided a panel of IHC images staining with stem/progenitor cell markers including CD44, TP63, and LGR5 (Jiang et al., *Oncotarget*, 7, 76159-68, 2016), on prostatic organoids derived from $R26^{mTmG/hAR};Osr1^{Cre/+}$ mice (Supplementary Fig. 7d-f). Both antiandrogen and Wnt inhibitors appear to inhibit the expression of the above markers.

Reviewer #3 (Remarks to the Author):

*In this manuscript, using a murine prostate cancer model $R26^{mTmG/hAR};Osr1Cre/+$ and single cell RNA-seq technique, the author tried to explore the mechanism of AR-driving prostate cancer. They identified that *Igfl* and *Wnt* signaling pathway were upregulated at high-grade prostatic intraepithelial neoplasia (HGPIN) and invasive prostatic adenocarcinoma (PCa) lesions in mice. They claimed that *Igfl* and *Wnt* signaling were the downstream targets of AR. Finally, to treat the murine PCa, they administrated the organoids which derived from PCa cells of $R26^{mTmG/hAR};Osr1Cre/+$ with WNT inhibitor. Overall, the concept, animal model and therapeutic strategy in the study are not novel, although the single cell RNA-seq data from mice are beautiful and solid. Many concepts have been reported. I also didn't see many human relevant data in the manuscript.*

In fact, whereas the AR mediated signaling pathways has been implicated in prostate tumorigenesis for decades, the fundamental mechanisms underlying AR functioning as a tumor promoter in inducing prostatic oncogenesis still remain elusive in the field. In this study, we directly addressed this important and long-standing question. Using the new and biologically relevant *in vivo* model, scRNA-seq analyses, and other experimental approaches, we identified that aberrant AR activation in prostatic Osr1-lineage cells elevates the IGF1 axis to initiates oncogenic transformation and PIN development, and elevated IGF1 signaling further activates Wnt/ β -catenin pathways to promote PIN to progress to prostate tumor. Specifically, data from scRNA-seq analyses provide new and high-resolution depiction for oncogenic AR action in prostate oncogenesis at single cell resolution, which has never been reported in the field. In addition, the AR transgenic mice used in this study are very different from previous prostate cancer mouse models regulated by the mouse probasin (PB) promoter. Therefore, data generated from these new, unique, and relevant experimental models and approaches provide fresh and significant insight into our current understanding of androgen signaling and prostate tumorigenesis. We have also thoroughly gone through the Reviewer's comments and addressed each of them below.

It seems this murine prostate cancer model ($R26^{mTmG/hAR};Osr1Cre/+$) is more relevant to human PCa

*compared with Pten-knock out murine PCa model. Could the author comment this? Is $R26^{mTmG/hAR}:Osr1Cre/+$ mice PCa model resistant to castration/AR antagonist? If so, the author could focus on its mechanism. If not, the author could establish a castration/AR antagonist resistant model using $R26^{mTmG/hAR}:Osr1Cre/+$ model. Another question is that I don't understand why the author select WNT inhibitor to treat the mice PCa in this study. According to the results from this study, *Igf1* and *Wnt* signaling are the AR downstream targets. If so, treatment with AR antagonist may be enough to inhibit these signaling pathways and the progression of PCa. WNT signaling pathway is broad in the body, so the treatment with WNT inhibitors may have more side effects than AR inhibitors.*

Unlike other human malignancies, the AR and androgen signaling pathways are essential for prostate tumorigenesis. Therefore, as stated above, it has been extremely important to develop the biologically relevant *in vivo* models that can recapitulate oncogenic AR action in prostate cancer development and progression. The $R26^{mTmG/hAR}:Osr1Cre/+$ mice used in this study developed high-grade PIN and invasive prostatic adenocarcinoma lesions, mimicking what occurs in human prostate cancer. To gain more and in-depth insight into this mouse model, in this study we performed a series of experiments to assess the regulatory role of AR in prostate tumorigenesis. As detailed in this study, we demonstrate the new mechanism by which transgenic AR activates IGF1 and Wnt signaling in prostatic *Osr1*-lineage cells to initiate oncogenesis and induce PIN and tumor development. Given that aberrant activation of Wnt signaling pathways has been frequently observed in advanced human prostate cancer, implicating its potential role in tumor progression (Robinson et al., *Cell*, 161, 1215-28, 2015), we then tested if Wnt activation can hijack androgen signaling to enhance tumor growth and progression using both organoid cultures and *in vivo* tissue graft experiments. As shown in the current revised manuscript, we observed more potent inhibitory effects on tumor growth by co-targeting androgen and Wnt signaling than targeting each pathway alone. We also appreciate the Reviewer's comments regarding the Wnt targeted therapies and have provided more rationale for the designed experiments in the current revision.

*1. For the single cell RNA-seq experiments in Figure 3, it seems the control samples are not quite suitable. For example, the authors tried to explore the impact of AR overexpression in *Osr1*+ prostatic progenitor. In this case, the experimental group should be $R26^{mTmG/hAR}:Osr1Cre/+$ and control group should be $R26^{mTmG/+}:Osr1Cre/+$, which just have one variate (*hAR*). Then we can easily compare the gene expression in the mGFP positive cells between these two groups. However, they just separated *hARTg*-negative and *hARTg*-positive basal epithelial cells. These two types of basal epithelial cells may originally express different genes because they belong to *Osr*-negative and *Osr*-positive basal epithelium. The analysis in Figure 3 could mislead the study.*

We appreciate the Reviewer's comments. Actually, we have done the similar analyses as the reviewer suggested (see the attached "Figure for the Reviewer"). Interestingly, we observed that normal prostatic BE cells from the control mice were mainly clustered into the BE1 cluster, possessing very similar cellular properties as BE1 cells that were identified in $R26^{mTmG/hAR}:Osr1Cre/+$ samples. GSEA using the DEGs between total BE cells of $R26^{mTmG/hAR}:Osr1Cre/+$ versus control samples identified enriched IGF1 signaling. In addition, GSEA with DEGs between BE2 clusters of $R26^{mTmG/hAR}:Osr1Cre/+$ versus total BE cells of controls showed very similar enriched signaling pathways as identified in total BE cells of the control mice. Based on these data and the high-resolution provided by scRNA-seq analyses, we assessed the molecular characteristics induced by *hARTg* expression through the comparison between *hARTg*+ and *hARTg*- basal epithelial cells in the same microenvironment. These analyses aimed to identify cell-cell interactions between basal and luminal epithelial cells and stromal and epithelial cells during the course of tumor initiation and progression. Data from other different experimental approaches also fully support our scRNA-seq data, demonstrating a regulatory role of transgenic AR in inducing IGF1 and Wnt signaling in prostatic epithelial oncogenesis. In response to the Reviewer, we modified the related sections to make the scientific rationale and results clearer to the reviewer in the current revision (see the end of page 9 and the beginning of page 10).

2. Another question is how many mice for one group were used in the single cell RNA-seq? This is important because the difference between individuals may also impact the results.

We used two sets of mice for this analysis, and observed similar data. In addition, as detailed in the manuscript, we have used other different experimental approaches to validate our findings from sc-RNA-seq analyses.

3. I am very curious that *Osr1*-Cre-driven hARTg but not *PB*-Cre-driven hARTg transgenic mice can develop prostatic adenocarcinoma lesions. Why does PCa specifically occur in *Osr1*+ prostatic progenitor? What is the role of *Osr1* during the tumorigenesis? Does *AR* overexpression reprogram the transcriptome and epigenome in *Osr1*+ prostatic progenitor at early stage? On the other hand, *Osr1* not only expresses in prostatic progenitor but also expresses in other cells such as fibroblasts and smooth muscle cells (Fig. 1B-D), indicating the specific epigenetic or cellular environment in *Osr1*+ prostatic progenitor may be important for the oncogenesis function of *AR*. In other words, the other factors but not just *AR* and *Osr1* are also important for the progression of prostatic adenocarcinoma. The authors indicated that *AR* upregulated *Igf1* and *Wnt* signaling in prostatic progenitor, but why overexpression of *AR* didn't impact the same pathways in other *Osr1*+ cells.

Again, the Reviewer raised a very interesting and important point in prostate tumorigenesis. Our *in vivo* tracing experiments demonstrated that embryonic *Osr1*-expressing cells are able to differentiate into prostatic basal and luminal epithelial lineages in response to raising androgen levels during prostate prepubescent and pubertal development (see Fig. 1), suggesting their prostatic epithelial progenitor properties. Our co-IF analyses further showed the prostatic basal epithelial cell properties of *Osr1*-lineage cells in comparison to *PB*-lineage cells. Development of HGPIN and prostate tumors was only observed in *AR* transgenic mice driven by *Osr1*-Cre, but not *probasin*-Cre {Zhu et al., *J. Biol. Chem.*, 286, 33478-88, 2011}. Multiple lines of evidence have also shown prostatic basal epithelial cells are very plastic and possess prostatic epithelial progenitor properties {Saha et al., *Oncotarget*, 7, 25194-207, 2016; Ousset et al., *Nat. Cell Biol.*, 14, 1131-8, 2012; Toivanen et al., *Stem Cell Reports*, 6, 660-7, 2016}. These data all support the critical role of prostatic *Osr1*-lineage cells in prostatic tumor initiation. Additionally, we do not know the exact reasons why only prostatic epithelial rather than mesenchymal/stromal oncogenesis appeared in *R26^{mTmG/hAR};Osr1^{Cre/+}* mice. There are many factors that may contribute to the phenotypes. For example, it has been shown that epithelial and stromal cells have different proliferative potentials. Interestingly data from our gene-gene correlation analyses showed the significant association between the expression of *Osr1* and other stem/progenitor markers in prostatic epithelial cells rather than stromal cells, suggesting the different cellular properties of epithelial and stromal *Osr1*-lineage cells in the prostate. Nonetheless, these interesting questions may only be resolved through further investigations using this new and relevant mouse model.

4. Although the author showed that *AR* can directly bind to the chromatin at *Igf1* loci, we still don't know if hAR can directly regulate the transcriptional level of *Igf1*. *AR* has about 60,000 binding sites at chromatin, but *AR* does not regulate the transcriptional level for every gene which has *AR* binding sites. To confirm if *AR* regulates the transcriptional level of *Igf1* or *Wnt* signaling, more experiments such as treatment the organoid with *AR* agonist/antagonist (DHT/enzalutamide) and qRT-PCR should be done. The same experiments in mice should be confirmed in human prostate cancer organoids or cell lines.

In response to the review, we examined the expression of IGF1R in organoids derived from hARTg+ *Osr1*-lineage cells. Results from qRT-PCR experiments showed increased expression of *Igf1r* transcript in the samples treated with DHT in comparison to those either without DHT or treated with antiandrogen, Enz (Supplementary Fig. 5c). We also examined the expression of IGF1R in several human prostate cancer cell lines, and observed the upregulation of IGF1R transcripts only in *AR* positive cancer cell lines, which is similar to the early study (Pandini et al., *Cancer Res.*, 65, 1849-57, 2005). Thus, we did not provide those duplicated data here.

5. In Figure 1H2 and I2, *Osr1* is a transcript factor, but why does most of *Osr1* locate at membrane? Could the author comment this?

In response to the reviewer, we provided new data in the above set of experiments with the different *Osr1* antibody (Supplementary Table 9). Both nuclear and cytoplasmic staining were observed in the current revised images (see Fig. 1h2, Fig. 1i2, and Supplementary Fig. 7j).

6. Does *Osr1* specifically expressed in prostate prostatic epithelial cell? If not, we may see hAR expression in other type of cells such as fibroblasts and smooth muscle cells. Could the author comment this?

We appreciate the Reviewer's insightful point. In our sc-RNA-seq analyses, we also observed *Osr1* expression in smooth muscle cells and fibroblasts in mouse prostates, especially in embryonic tissues. However, the gene-gene correlation analyses showed the significant association between *Osr1* and other stem/progenitor marker expression in prostatic epithelial rather than stromal cells, suggesting the different cellular properties and role of *Osr1*-expressing cells in epithelial and stromal compartments during prostate development. Moreover, hARtg expression driven by *Osr1*-Cre mainly revealed in prostatic epithelia of $R26^{mTmG/hAR}:Osr1^{Cre/+}$ mice, further implicating that prostatic epithelial *Osr1*-lineage cells possess differentiation and expansion ability during prostate development. In the current revision, we also modified related text to address the related questions.

7. “TP63+mGFP+ basal epithelial cells were only observed in prostate tissues of $R26^{mTmG/+}:Osr1^{Cre/+}$ mice but not in those of $R26^{mTmG/+}:PBCre/+$ mice (Supplementary Fig.S3G1-1' versus S3G2-2)”. What is the role of *Osr1*+ basal epithelial cells? Does *Osr1*+ basal epithelial cells mainly drive oncogenesis and tumor development in this model?

In this study, using different experimental approaches, we identified the subpopulation of prostatic basal epithelial cells showing both TP63+ and mGFP+ in *Osr1*-Cre but not *PB*-Cre mice. scRNAseq data also showed hARtg+ BE2 clusters composing different transcriptomic profiles even in merged data between control and $R26^{mTmG/hAR}:Osr1^{Cre/+}$ mice. It has been shown that prostatic basal cells are very plastic and possess prostate progenitor properties {Saha et al., *Oncotarget*, 7, 25194-207, 2016; Ousset et al., *Nat. Cell Biol.*, 14, 1131-8, 2012; Toivanen et al., *Stem Cell Reports*, 6, 660-7, 2016}. Additionally, given the previous data showing that only *Osr1*-Cre but not *probasin*-Cre driven hAR leads to prostate tumorigenesis in mice, our findings indicate the critical role of prostatic *Osr1*-Cre lineage cells in initiating prostate tumorigenesis. Again, we greatly appreciate the Reviewer's insightful points and believe they could only be resolved through more investigation.

8. In the abstract, the author mentioned human prostate cancer, but I didn't see many human relevant data in this manuscript. Does AR overexpression upregulate IGF1 and WNT signaling in human prostate cancer?

As described earlier, we have provided more clinical evidence to demonstrate the regulator role of AR on IGF1 and Wnt signaling pathways in human prostate tumorigenesis in the current revision. They include 1) the co-occurrence of alterations and mutations of *AR* and *IGF1R*, *CTNNB1*, or *MYC* genes and their expression in both primary and advanced tumor samples (Fig. 7f), 2) a significant correlation between those abnormalities in the above samples (Fig. 7g), 3) a significantly positive correlation between the transcripts of *AR* with *IGF1R*, *CTNNB1*, or *MYC* in primary PCa samples (Fig. 7h), and 4) an enrichment of AR on the *IGF1R* gene locus in ChIP-seq datasets of human PCa samples in comparison to controls with normal prostate tissues (Supplementary Fig. 6).

REVIEWER COMMENTS

Reviewer #1 (Remarks to the Author):

The manuscript has been dramatically improved by the revisions.

Some very minor changes (e.g. addition of statistical tests, details about computational methods, and details about control mice).

the authors should add statistics next when it is stated significant.

I appreciate the addition of FDR values to Fig 5C, but "FDR=0.000" should be changed to "FDR < f", where f should be replaced by the smallest non-zero FDR value that could be calculated (since the actual FDR value is probably just too small to calculate rather than actually being zero).

Spelling error: "conduced"  "conducted" (p15, L308)

Typo in Fig 7d: I think 7d was meant to be hARtg-, not hARtg+.

Reviewer #2 (Remarks to the Author):

The results of the revised paper are clinically relevant. Most comments of myself in the first round review were related to that. I am satisfied with additions of data and improved discussions. The authors carefully revised the manuscript.

Reviewer #3 (Remarks to the Author):

The authors have done a good job in addressing my comments.

Reviewer #4 (Remarks to the Author):

I only have some suggestions on the statistical analyses in this paper.

1. Regarding the pseudotime/trajectory inference questioned by Reviewer 1, I would suggest the authors perform the analysis using Slingshot in addition to Monocle 2. The reason is that several studies found Slingshot to give less noisy and thus more interpretable pseudotime trajectories. I would also like to suggest the authors consider the uncertainty in pseudotime inference and its effect on the identification of differentially expressed genes. For example, the method PseudotimeDE offers a way to account for this uncertainty in identifying genes with changing expression along one pseudotime branch.

2. Regarding the differential gene expression analysis, it was reported that edgeR may have inflated false discovery rates if its negative binomial distributional assumption does not hold. I would suggest that the authors add limma-voom, which has a different parametric model assumption, as an alternative if the sample size is not greater than 8 per condition (otherwise, Wilcoxon would become applicable). Another option is that the authors may use Clipper, a nonparametric method designed for small sample sizes, with logCPM values as the input.

I hope these two suggestions can help strengthen the statistical rigor in terms of controlling false positives.

Reviewer #1 (Remarks to the Author): *The manuscript has been dramatically improved by the revisions. Some very minor changes (e.g. addition of statistical tests, details about computational methods, and details about control mice). The authors should add statistics next when it is stated significant. I appreciate the addition of FDR values to Fig 5C, but “FDR=0.000” should be changed to “FDR < f”, where f should be replaced by the smallest non-zero FDR value that could be calculated (since the actual FDR value is probably just too small to calculate rather than actually being zero).*

We greatly appreciate the Reviewer’s points and have carefully gone through the manuscript, making sure detailed information for statistical tests, computational methods, and control mice were clearly placed in the text (see changes in red ink in the current revised manuscript) and in related figures and figure legends as well. Specifically, we have made the exact changes as suggested by the Reviewer in the revised Fig. 5.

Spelling error: “conduced”  “conducted” (p15, L308).

The above spelling error has been corrected (see page15, L312)

Typo in Fig 7d: I think 7d was meant to be hARtg-, not hARtg+.

The above typo has been corrected in the current Fig. 7d.

Reviewer #4 (Remarks to the Author): *I only have some suggestions on the statistical analyses in this paper.*

1. Regarding the pseudotime/trajectory inference questioned by Reviewer 1, I would suggest the authors perform the analysis using Slingshot in addition to Monocle 2. The reason is that several studies found Slingshot to give less noisy and thus more interpretable pseudotime trajectories. I would also like to suggest the authors consider the uncertainty in pseudotime inference and its effect on the identification of differentially expressed genes. For example, the method PseudotimeDE offers a way to account for this uncertainty in identifying genes with changing expression along one pseudotime branch.

In response to the Reviewer’s suggestion, we performed Slingshot analyses to characterize the cell fates of hARtg+ and hARtg- cells through PIN initiation and progression to prostatic adenocarcinomas. As shown in Supplementary Figure 8a-b, hARtg+ epithelial cells from merged PIN and PCa samples were derived from BE2 cluster, a PIN cluster, and differentiated to Curve 1 mainly possessing LE1 and LE2, two tumor clusters. In contrast, hARtg- cells were derived from BE1 cells and differentiated to curves mainly containing normal LE clusters 5-7. These data are very similar to our results generated using Monocle2. In addition, using the PseudotimeDE package as suggested by the Reviewer, we also addressed pseudotime inference uncertainty of differentially expressed genes on different branches. As presented in Supplementary Figure 8c, we observed significant correlations of changes in the expression of hARtg and Wnt downstream targets.

2. Regarding the differential gene expression analysis, it was reported that edgeR may have inflated false discovery rates if its negative binomial distributional assumption does not hold. I would suggest that the authors add limma-voom, which has a different parametric model assumption, as an alternative if the sample size is not greater than 8 per condition (otherwise, Wilcoxon would become applicable). Another option is that the authors may use Clipper, a nonparametric method designed for small sample sizes, with logCPM values as the input. I hope these two suggestions can help strengthen the statistical rigor in terms of controlling false positives.

We agree with the Reviewer’s point. Actually, we have been very cautious with our analyses processed with edgeR. In our current data presented in Figure 5, we had filtered out genes that did not fit negative binomial distributional assumption based on zero inflated data and assess any significant violation of negative binomial assumption of the data such as inflated false discovery rates. As shown in the attached plots below, our data fit well to negative binomial (blue line) compared to Poisson (black line). Additionally, in response to the Reviewer, we also performed Clipper analyses using the above datasets, and included the data in the revised Supplementary Tables 5-7. In the analysis, we identified 66.6, 84.3, and 99.2 % of DEGs that were identified from PIN vs WT, PCa vs WT, and PCa vs PIN, respectively, by edgeR to overlay with our data generated using Clipper. We modified the related text in “Methods” of the current revision.

REVIEWERS' COMMENTS

Reviewer #4 (Remarks to the Author):

The authors have successfully address my comments.